# A Generative AI Framework for Cognitive Intervention in Older Adults: An Integrated Engineering Design and Clinical Protocol

**DOI:** 10.3390/healthcare13243225

**Published:** 2025-12-10

**Authors:** Taeksoo Jeong, Geonhwi Hwang, Doo Young Kim

**Affiliations:** 1Department of Rehabilitation Medicine, International St. Mary’s Hospital, Catholic Kwandong University College of Medicine, Incheon 22711, Republic of Korea; 2Department of Applied Physics, Graduate School of Engineering, Hokkaido University, Sapporo 060-8628, Japan; 3The Convergence Institute of Healthcare & Medical Science, Catholic Kwandong University College of Medicine, Incheon 22711, Republic of Korea

**Keywords:** cognitive intervention, cognitive training, older adults, generative AI, multi-agent system, digital health, aging society

## Abstract

**Background:** Digital exclusion is a validated risk factor for cognitive decline in older adults. Digital interventions exhibit high dropout rates due to low digital literacy, technology anxiety, and limited adaptation to individual states, resulting in limited real-world transfer. **Objective:** This protocol aims to present the CTC Framework (Coach–Teacher–Companion), a tri-agent generative AI system proposed for exploring the feasibility of adaptive cognitive interventions in older adults with existing digital access. The protocol provides technical architecture, feasibility-stage implementation procedures, and methodological and ethical guidelines to assist clinicians in safely applying AI-based cognitive interventions in clinical research settings. **Methods:** The framework integrates three AI agents (Coach, Teacher, and Companion) designed to provide behavioral, cognitive, and emotional support. The system is designed to embed cognitive exercises in daily activities, monitor emotional states, and incorporate accessibility features for age-related limitations. Implementation safeguards include digital literacy assessment (MDPQ-16), technology anxiety monitoring (CARS), emotional safety protocols, and data privacy protections. The protocol specifies a six-week feasibility study (*n*
=14, MMSE 18–25) to evaluate usability (System Usability Scale, primary outcome), user experience (UEQ-S), psychological needs satisfaction (BPNS), emotional safety (PANAS), adherence, and preliminary cognitive outcomes (MMSE, TMT-A/B, Digit Span). **Conclusions:** The CTC Framework is designed to provide methodological and ethical safeguards for clinical implementation, including standardized procedures for digital literacy assessment, technology anxiety management, emotional safety monitoring, and data privacy protections. Empirical validation of the framework’s feasibility and efficacy is required through future studies.

## 1. Introduction

Digital exclusion has emerged as a significant risk factor for cognitive decline in older adults. Large-scale studies involving approximately 24,000 participants show that digitally excluded older adults face a 1.4- to 2.04-fold higher risk of developing mild cognitive impairment (MCI) [1]. This creates a paradox: those who might benefit most from digital cognitive interventions are least likely to engage due to low digital literacy, technology anxiety, and distrust [2]. Conversely, a 12-year cohort study demonstrated that sustained digital engagement increased the probability of recovery from MCI to normal cognition by sixfold to ninefold [3]. However, achieving such engagement requires systems capable of adapting to users’ cognitive, emotional, and behavioral states [4].

Current digital cognitive interventions face critical barriers, including low digital literacy, technology anxiety [5], static training, and limited personalization. Addressing these barriers requires systems supporting self-efficacy [6] and intrinsic motivation through Self-Determination Theory principles [7].

Large language models (LLMs), a form of artificial intelligence (AI), demonstrate capacity for adaptive interactions addressing the cognitive and social needs of older adults [8,9]. However, clinical implementation requires methodological frameworks addressing digital literacy assessment, cognitive load management, emotional safety monitoring, and data privacy protections.

In this paper, we introduce the CTC Framework (Coach–Teacher–Companion), a tri-agent system designed to explore reducing interaction barriers among older adults with baseline digital access. The Coach is designed to support adherence through daily activity integration, the Teacher provides adaptive cognitive training, and the Companion monitors emotional states through voice prosody and interaction patterns. The framework employs accessibility-focused interface design, adaptive difficulty adjustment, and emotionally supportive interaction, with sustained internet connectivity and device availability as prerequisites within resource-constrained but digitally accessible settings.

The protocol aims to establish an implementation methodology for AI-based cognitive interventions in older adults. Specific objectives include specifying participant screening procedures (digital literacy assessment, technology anxiety monitoring), defining safety protocols (cognitive load thresholds, discontinuation criteria), establishing data privacy safeguards, and designing a feasibility study to evaluate usability and preliminary outcomes. This protocol provides clinicians and researchers with standardized procedures for safely implementing AI-based interventions in clinical research settings.

## 2. Background and Theoretical Framework

### 2.1. Digital Exclusion and Cognitive Health: The Central Problem

Digital exclusion and cognitive impairment interact through reduced mental stimulation and amplified social disconnection, creating a significant public health challenge in aging populations. Digital exclusion particularly affects processing speed, an early indicator of cognitive deterioration [3]. Social frailty mediates this relationship, with odds ratios of 1.30–1.41 for predicting cognitive impairment [10].

Conversely, sustained digital engagement protects against decline, with 12-year cohort data showing sixfold to ninefold increased probability of MCI recovery [3]. Protective effects operate through cognitive stimulation and enhanced social connectivity [11,12].

Reducing digital exclusion requires adaptive technology, engagement theory, and safety protocols. Section 2.2, Section 2.3 and Section 2.4 examine how generative AI, Self-Determination Theory, and clinical implementation address these requirements.

### 2.2. Generative AI as a Technological Pathway

Generative AI enables adaptive interactions through multimodal processing of linguistic, behavioral, and emotional signals [13,14,15]. Emotional recognition may reduce barriers to digital participation by mitigating technology anxiety [5] and providing context-sensitive support for older adults [16].

AI companion systems demonstrate sustained engagement exceeding 30 daily interactions and 95% reduction in loneliness [17], with effectiveness mediated by perceived humanness and structured guidance [18,19]. The CTC Framework extends this approach by integrating behavioral coaching and cognitive training alongside emotional support.

Implementation challenges include conversational adaptation (verbosity, tone, and complexity) [15,20], hallucination risks in vulnerable populations [21], and potential over-dependence requiring balance with human interaction [16,22]. Section 5 specifies corresponding safeguards.

### 2.3. Theoretical Basis: Self-Determination Theory and Sustained Digital Engagement

Sustained engagement requires addressing psychological barriers. Self-Determination Theory (SDT) posits that intrinsic motivation depends on satisfying three psychological needs: autonomy, competence, and relatedness. Satisfying these needs promotes behavior internalization and sustained engagement [7]. Figure 1 illustrates the SDT framework for AI-based cognitive training applications.

These three psychological needs also relate to technology anxiety and self-efficacy, key factors in digital exclusion. The CTC Framework adopts SDT as a theoretical basis, proposing design features intended to support the three psychological needs. Autonomy support is proposed through user-configurable task timing and agent interaction preferences (Section 3.3.2), intended to provide users with control over intervention parameters. Competence support is proposed through adaptive difficulty targeting 85% success rates (Section 3.2.2), intended to maintain optimal challenge while preventing frustration. Relatedness support is proposed through emotionally responsive dialogue that acknowledges user affect (Section 3.2.1), intended to foster perceived social connection. The feasibility study employs the Basic Psychological Needs Satisfaction Scale (BPNS) to explore whether these theoretically motivated design features correspond to participants’ perceived need satisfaction (Section 5.4.3). This exploratory assessment will inform whether the proposed design-to-need mappings warrant further investigation in efficacy trials.

### 2.4. From Requirements to Framework Design

Many cognitive training applications employ static interfaces with limited adaptation [23,24], rule-based chatbots limiting emotional flexibility, and gamification strategies yielding short-term engagement [25,26], while inadequate accommodation of age-related changes induces cognitive overload [27].

The CTC Framework is designed to address these limitations through generative AI-based adaptation, interface design minimizing cognitive load, engagement strategies grounded in SDT, and implementation safeguards for digital literacy, emotional safety, and data privacy. Section 3 details the technical architecture, while Section 5 provides clinical deployment protocols.

## 3. System Architecture and Design Principles

This section presents the technical architecture operationalizing the principles described in Section 2. The system design addresses multiple dimensions relevant to clinical implementation: embedding cognitive training in daily activities, fostering social connection, and ensuring accessibility for digitally vulnerable older adults.

### 3.1. Design Objectives

The system design follows three objectives: embedding cognitive training within natural daily activities to ensure ecological validity and sustained engagement, fostering social connection and emotional stability to combat isolation and cognitive decline, and ensuring accessibility for digitally vulnerable older adults. These objectives inform the technical architecture (Section 3.2) and interface design (Section 3.3).

#### 3.1.1. Encouraging Natural Daily Activities

Embedding cognitive training in daily activities enhances ecological validity and functional transfer by integrating cognitive stimuli into goal-oriented contexts. The system is designed to activate cognitive functions through everyday activities such as preparing meals, managing medications, or grocery shopping.

Generative AI dynamically generates prompts aligned with individual habits and schedules, reframing cognitive exercises as natural extensions of daily activities (e.g., recalling grocery items for memory training, embedding attention-shifting tasks in navigation). Such contextualization may foster intrinsic motivation and digital self-efficacy, contributing to sustained engagement [28]. Real-world task contextualization improved usability and engagement among older adults with MCI compared to abstract exercises [29], enhanced adherence and cognitive transfer [30], and demonstrated superior long-term retention when aligned with personal habits [31,32,33].

#### 3.1.2. Promoting Social Connection and Emotional Stability

Social isolation (often exacerbated by aging, chronic illness, family estrangement, and limited digital accessibility) accelerates cognitive decline, with socially isolated older adults facing 1.4 to 2.04 times higher risk of developing MCI [1,34].

The CTC Framework employs three approaches to foster social connection and emotional stability. First, the AI agent responds to emotional cues through empathetic dialogue with clearly defined personas [35], consistent with older adults’ preference for transparent acknowledgment of AI limitations [6]. Second, the system enhances personalized dialogue by embedding past photos, family stories, and meaningful memories into conversational flow, inducing reminiscence therapy-like effects. Photo-based storytelling systems have achieved 45% higher daily usage rates compared to generic content, with significant improvements in both cognitive function and emotional well-being [36]. Third, the system enables caregivers, family members, and healthcare professionals to monitor user status and provide feedback, supporting collaborative care [37].

AI companion systems such as ElliQ demonstrate adaptive engagement with older adults, with 80% of users reporting reduced loneliness [17]. To address concerns about control, a barrier to adoption, the system clarifies AI’s supportive role through transparent interactions [6].

#### 3.1.3. Ensuring Accessibility for Digitally Vulnerable Older Adults

Despite increased digital participation, barriers remain: emotional resistance, cognitive overload, and low digital self-efficacy hinder sustained engagement [5]. Cross-cultural research in Germany, Japan, and Thailand identified common obstacles including limited digital literacy, psychological resistance, and infrastructural gaps [38]. Conventional “senior-friendly” applications often rely on large fonts or simplified layouts without addressing psychological and cognitive barriers to digital interaction. Technology anxiety has been shown to indirectly influence the intention to use digital public services through perceived usefulness and self-efficacy, emphasizing the need for strategies that alleviate anxiety and enhance accessibility [5].

The system employs four strategies: First, a multimodal interface with large-font text, icon-based menus, and voice guidance minimizes cognitive confusion through consistent positioning of visually distinguishable interactive elements [3]. Second, behavioral scaffolding facilitates gradual skill acquisition through dynamic prompts and responsive feedback, enabling users to build successful interaction habits over time [39]. Third, the AI agent provides emotionally supportive feedback (e.g., “You are doing well”), supporting self-efficacy development [40]. Fourth, a family caregiver feedback loop enables indirect support, with family members receiving periodic summaries of usage patterns to provide encouragement or intervene when necessary, as family involvement positively influences older adults’ continued engagement with digital activities [41]. Economic accessibility is an important consideration, particularly in regions with high poverty rates among older adults. Web-based deployment on existing devices without specialized hardware is designed to support accessibility in low-income populations [42,43]. These considerations inform digital literacy screening and technology anxiety assessment protocols in Section 5.

### 3.2. System Architecture

The system integrates generative AI, cloud infrastructure, and user-centric interfaces for personalized cognitive intervention, providing technical specifications for clinical deployment. The architecture comprises three core components: an AI agent powered by large language models, a cloud-based data management system, and adaptive user interfaces optimized for accessibility.

#### 3.2.1. Modular Agent Design

The system employs a the modular agent architecture with three distinct personas supporting functional specialization and emotional regulation. Excessive agent fragmentation can increase cognitive burden and impair usability [44]. Cognitive load theory suggests that clear, distinct categories improve information processing compared to ambiguous or overlapping functions. Studies demonstrate that simpler, well-defined personas reduce cognitive flexibility load and foster clarity, trust, and emotional engagement [45].

Research suggests that 2–4 distinct personas improve task completion compared to single-agent or many-agent configurations [46], aligning with working memory constraints in older adults [47]. The optimal agent configuration for this population requires empirical validation through the feasibility study. The system employs three personas (agents) as detailed in Table 1, whose structure is designed to reduce cognitive load and to offer a functional taxonomy for comparable interventions.

Figure 2 shows modular agent architecture of the CTC Framework. The Coach Agent reinforces executive function through behavioral interventions integrated into daily activities. Digital coaching interventions have demonstrated effects on physical activity and medication adherence in older populations [48,49]. The agent employs implementation intention strategies and contextual reminders to support adherence [50].

The Teacher Agent focuses on direct cognitive intervention through structured exercises embedded in daily contexts. AI-assisted cognitive training has shown improvements in global cognition and executive function among older adults with MCI [51,52]. The agent employs spaced repetition algorithms and adaptive difficulty adjustment [53].

The Companion Agent addresses emotional well-being’s critical role in cognitive health. Conversational AI companions can alleviate loneliness and depressive symptoms among isolated older adults through daily interaction and emotional support [54,55]. This agent employs reminiscence therapy principles [56]. Longitudinal studies suggest that emotionally supportive AI interactions may increase self-disclosure and strengthen user trust [57].

This structure supports functional specialization while limiting cognitive load. Each agent maintains distinct visual and auditory characteristics to facilitate recognition and reduce confusion [58]. Users or caregivers can configure agent use based on needs. These principles inform participant onboarding and agent interaction monitoring in Section 5.

#### 3.2.2. Context-Adaptive Response System

The CTC Framework is designed to employ a context-adaptive processing pipeline that adjusts to user emotional state, behavioral patterns, and cognitive capacity. The pipeline comprises four stages [59,60].

**Stage I: Multimodal Data Sensing.** The system captures user interactions through behavioral, vocal, performance, and temporal features. Each persona agent (Coach, Teacher, and Companion) prioritizes relevant features to reduce computational load [61,62].

**Stage II: Persona-Specific Context Recognition.** A cognitive load metric based on workload assessment frameworks is proposed to quantify user state [63]: (1)$$L_{\mathrm{cog}}=w_{1}\cdot \frac{\Delta t_{\mathrm{resp}}}{{\bar{t}}_{\mathrm{base}}}+w_{2}\cdot e_{\mathrm{rate}}+w_{3}\cdot \sigma_{\mathrm{att}}^{2}$$
where $\Delta t_{\mathrm{resp}}/{\bar{t}}_{\mathrm{base}}$ represents normalized response time deviation, $e_{\mathrm{rate}}$ captures error frequency, and $\sigma_{\mathrm{att}}^{2}$ measures attention variance. Initial weight parameters ($w_{1}=0.4$, $w_{2}=0.35$, $w_{3}=0.25$) are theory-derived proposals prioritizing processing speed as the primary indicator, consistent with its established role as the primary mediator of age-related cognitive performance [64]. The metric is constrained to $L_{\mathrm{cog}}\in \left[0,2\right]$. The feasibility study will employ this fixed parameter set to assess system usability and identify necessary parameter refinements for subsequent trials. The Companion agent employs valence-arousal mapping for emotional state assessment [65,66].

**Stage III: Emotionally Adaptive Response Strategy.** The Teacher agent is designed to employ an adaptive difficulty system based on item response theory: (2)$$P_{t}=\frac{1}{1+\exp\{-a\left(\theta_{t}-b_{t}\right)\}},\;b_{t+1}={\mathrm{clip}}_{\left[b_{\min},b_{\max}\right]}\left(b_{t}+\eta \left(P_{t}-P^{*}\right)\right)$$
where $\theta_{t}$ represents the estimated user ability, and $b_{t}$ is the task difficulty. The system targets success probability $P^{*}$ at approximately 0.85, based on optimal challenge principles [67]. This target represents a theory-derived design choice; the feasibility study will employ this fixed parameter to assess usability and participant response patterns, informing necessary adjustments for subsequent trials. The difficulty state is shared with the Coach and Companion agents to adjust emotional support and motivational strategies.

Response timing accommodates age-related processing speed changes [68].

**Stage IV: Feedback and Iterative Adaptation.** The system accommodates day-to-day variability: (3)$$\theta_{t+1}=\lambda \theta_{t}+\left(1-\lambda \right){\hat{\theta }}_{t}$$
where ${\hat{\theta }}_{t}$ is the maximum likelihood estimate based on recent performance and λ balances historical stability with recent responsiveness, accounting for day-to-day variability in cognitive aging [58].

#### 3.2.3. Technical Implementation

The technical implementation employs a triple-engine architecture that instantiates the four-stage processing pipeline through natural language processing components. This architecture supports real-time adaptation on resource-constrained devices [69]. Response timing adapts to individual processing speeds [70].

**Stage I: Perception Engine.** The Perception Engine implements multimodal data sensing through parallel processing streams. For speech input, the system employs a Whisper-based speech recognition model fine-tuned on older adult speech patterns [71,72]. Behavioral signals are captured through event detection by measuring touch duration, interaction attempts, and hesitation patterns [73].

**Stage II: Context Hub.** The Context Hub implements persona-specific recognition using a language model. The model maintains conversation context through a sliding window of 5 recent utterances, with 40% size reduction and 97% performance retention through distillation [74,75]. Intent classification employs multi-task learning across cognitive stimulation, emotional support, and activity-related intents [76]. Cognitive state estimation combines linguistic complexity measures with behavioral signals [77].

**Stage III: Response Engine.** The Response Engine employs cloud-based Application Programming Interface (API) access to LLMs (GPT-4o-mini, Gemini 1.5 Flash, or equivalent models) without fine-tuning. Each agent (Coach, Teacher, Companion) uses persona-specific prefix modules requiring <0.1% additional parameters [78]. The API-based architecture ensures user data is not retained for model training, consistent with data privacy requirements. Response generation incorporates safety constraints for age-appropriate content [79]. Implementation code is available (https://github.com/jeongtaeksoo/context-adaptive-cognitive-flow (accessed on 7 December 2025)).

The prototype architecture targets a response time under 3 s, intended to approach acceptable latency for conversational interaction [80]. The modular architecture is designed to support continuous adaptation [81] and persona switching between cognitive training, emotional support, and daily assistance modes [82]. Actual performance metrics will be obtained during pilot testing.

### 3.3. Interface and Interaction Design

Interface design is critical to addressing digital exclusion, as age-related changes in visual, motor, and cognitive function constitute primary barriers to technology adoption in older adults [27]. This section specifies accessibility parameters and interaction protocols implementing the framework’s commitment to inclusive design.

#### 3.3.1. UX/UI Design for Older Adults

Interface design for older adults accommodates age-related changes in visual acuity, cognitive processing, and motor control. Figure 3 illustrates the design principles across four key domains.

**Visual Design Parameters.** Text sizing (16–18 px, 1.5× line height) supports readability [84]. Contrast ratios (7:1 normal text, 4.5:1 large text) are guided by Web Content Accessibility Guidelines (WCAG) 2.1 AA standards [83].

**Navigation Architecture.** Working memory limitations in adults over 70 inform simplified architecture and reduced information load [47]. Navigation depth is limited to three levels [85]. Touch targets are 44 × 44 pixels minimum [86].

**Multimodal Interface.** Dual-coding combines icons with text labels. Skeuomorphic icons support comprehension [87].

**Behavioral Support Features.** Behavioral support includes scaffolding, progressive disclosure, and empathetic feedback.

#### 3.3.2. Task Scheduling and Adaptive Feedback

Task scheduling is designed to adapt to users’ daily activities and circadian rhythms, with cognitive tasks scheduled in the morning when cognitive performance peaks and social activities in the afternoon [88]. Response time allowances accommodate age-related processing speed changes [64].

Cognitive exercises embedded in daily activities (medication management, grocery shopping, appointment scheduling) show higher engagement and transfer effects than abstract training [89]. Performance monitoring uses composite indicators of accuracy, speed, and consistency. Reminders escalate from visual cues to auditory alerts (+15 min) to haptic feedback (+30 min) to accommodate varying levels of user attention.

#### 3.3.3. Conversational Design

Conversational tone impacts system acceptance and engagement. The system employs the Pleasure-Arousal-Dominance (PAD) model: moderate pleasure, lower arousal, and moderate dominance [90]. Gain-framed messaging (“Completing this task will help maintain your memory”) is employed [91].

Error messages avoid blame attribution (“Let’s try another way” rather than “That’s incorrect”) to maintain a positive user experience. Positive reinforcement and adaptive feedback support motivation [92]. Contextual continuity across sessions supports relationship quality and trust in long-term interactions [93], which may be particularly beneficial for users with MCI who rely on consistent interpersonal cues.

These interface design principles establish technical specifications for implementation. Section 4 describes a prototype demonstrating technical feasibility, while Section 5 provides clinical deployment protocols addressing participant safety, digital literacy, emotional monitoring, and data privacy.

## 4. Implementation Framework

The CTC Framework was implemented as a digital application. Agents coordinate responses: the Teacher adjusts task difficulty based on emotional assessment, while the Coach embeds activities within daily routines. This implementation demonstrates technical feasibility and provides reference code for clinical deployment.

### 4.1. Implementation Overview

The prototype was designed to instantiate the four-stage processing pipeline described in Section 3.2.2. Agents were designed to share contextual information for coordinated responses: when emotional distress is detected, the Teacher adjusts task difficulty (targeting 85% success rate) while the Coach provides motivational feedback and the Companion offers empathetic responses. The interface was designed to implement accessibility features specified in Section 3.3.1.

### 4.2. Operational Demonstration

Agent collaboration is illustrated through simulated scenarios demonstrating intended system behavior: *Morning engagement*—The Coach reviews scheduled activities; if low emotional valence is detected, the Teacher begins with easier exercises while the Coach suggests mood-lifting activities. *Cognitive training*—The Teacher monitors performance during memory exercises; when error rates increase, the Companion provides encouragement while the Coach suggests a break. *Evening reminiscence*—The Companion initiates conversation based on daily emotional patterns, with the Teacher embedding cognitive exercises and the Coach recording insights. Clinical deployment should monitor agent switching frequency, response appropriateness, and emotional trajectory.

### 4.3. Feasibility Demonstration and Validation Requirements

This prototype demonstrates the computational architecture for: (1) inter-agent communication, (2) context sharing across cognitive, emotional, and behavioral dimensions, and (3) coordinated response generation. Empirical evidence of technical feasibility will be obtained through the pilot study described in Section 5.

## 5. Methods

This section presents a feasibility protocol operationalizing the technical architecture (Section 3) through clinical procedures addressing participant screening, intervention delivery, outcome assessment, and safety monitoring.

The included six-week pilot phase evaluates three domains: (1) technical feasibility and adherence, (2) usability and cognitive engagement, and (3) emotional safety and well-being. Figure 4 illustrates study procedures and participant flow.

### 5.1. Study Design and Rationale

This pilot employs a single-arm, prospective design evaluating the feasibility and usability of a theory-informed prototype with fixed design parameters. Participants engage in AI-assisted cognitive sessions for six weeks (five sessions/week, 30 min/session). The study includes baseline assessment (T0), intervention phase with continuous digital monitoring, and post-intervention assessment (T1).

The primary objective is to assess feasibility through (1) adherence (≥70% session completion), (2) usability (SUS ≥ 70), and (3) emotional safety (negative affect within predefined thresholds). System parameters (cognitive load weights, adaptive difficulty thresholds, agent interaction protocols) are held constant to enable assessment of the prototype’s acceptability and safety; parameter optimization is planned for subsequent efficacy trials based on observed interaction patterns and participant feedback. The single-arm design follows recommendations for feasibility studies in vulnerable populations but precludes causal inference; observed changes cannot be attributed to the intervention without a control group. Future RCTs should compare the system with traditional cognitive training or single-agent programs.

### 5.2. Participants and Eligibility Criteria

#### 5.2.1. Inclusion Criteria

Participants are community-dwelling older adults aged ≥ 65 years with subjective cognitive decline or MCI (MMSE 18–25). Participants must have sufficient visual, auditory, and motor abilities to use a tablet or smartphone. Participants must provide informed consent and commit to the six-week intervention.

#### 5.2.2. Exclusion Criteria

Exclusion criteria: (1) moderate to severe dementia (MMSE < 18) or significant functional impairment, (2) major neurological or psychiatric disorders, (3) severe visual or hearing impairment interfering with device use, and (4) participation in another cognitive training study within three months. These criteria ensure safety while maintaining sample heterogeneity.

Digital literacy is assessed using the Mobile Device Proficiency Questionnaire (MDPQ-16) [94]. Technology anxiety is measured at baseline with the Computer Anxiety Rating Scale (CARS) [95], with scores > 2 SD above normative means triggering enhanced onboarding support. For participants meeting this threshold, weekly CARS assessments are administered during check-in calls to monitor anxiety trajectories and identify need for additional support or discontinuation. These baseline measures serve as covariates in usability analysis.

#### 5.2.3. Sample Size and Rationale

This pilot study employs a pragmatic sample size approach consistent with feasibility study guidelines [96]. Sample size must be justified to ensure meaningful estimation of feasibility indicators and intervention variability. Following established recommendations for single-arm pilot studies, approximately 12 participants are needed to achieve adequate precision in estimating variance and retention rates.

Given the single-arm design and practical constraints, a total of 14 participants will be enrolled, allowing for an anticipated 10% attrition while maintaining an adequate sample size for feasibility assessment. This sample size is consistent with pilot-study recommendations [96] and provides information to refine study procedures before progressing to a larger RCT. This sample size prioritizes feasibility assessment over hypothesis testing.

### 5.3. Intervention Protocol

Participants use a digital application integrating three AI agents (Coach, Teacher, and Companion). Each session includes the following:**Coach:** Behavioral guidance and activity scheduling.**Teacher:** Cognitive training embedded in daily activities.**Companion:** Emotional monitoring and empathetic feedback.

Implementation follows specifications in Section 3. Cognitive exercises include (1) episodic memory tasks (recalling grocery lists, medication schedules), (2) working memory tasks (backward digit span embedded in phone number entry), (3) attention tasks (identifying target items during simulated shopping), and (4) language tasks (word generation during recipe planning). Task difficulty is designed to adapt based on performance as specified in Section 3.2.2.

Participants complete 30 sessions over six weeks (five sessions/week, 30 min/session). This schedule is designed to generate sufficient interaction data (15 total hours) for feasibility assessment while minimizing participant burden, consistent with pilot study guidelines [97,98]. Standardized onboarding includes (1) device orientation checklist (approximately 15 min), (2) agent demonstration with scripted introduction (approximately 20 min), (3) supervised practice session (approximately 30 min), and (4) written manual provision. Detailed onboarding procedures are documented in the study protocol.

Following onboarding, participants complete all 30 intervention sessions independently using their personal devices at home, with research staff available via phone for technical support.

### 5.4. Outcome Measures and Assessment Schedule

#### 5.4.1. Feasibility Indicators

This feasibility study protocol specifies the evaluation of preliminary indicators for future trials. Feasibility will be assessed through (1) usability metrics (SUS score, with a threshold of ≥70 considered acceptable) and user satisfaction (UEQ-S), (2) preliminary cognitive function measures (MMSE, TMT-A/B, Digit Span), and (3) emotional state monitoring (PANAS) to ensure no sustained negative affect elevation.

#### 5.4.2. Primary Outcome

**System Usability Scale (SUS):** The SUS is a standardized 10-item scale [99] that measures overall usability and system efficiency. Each item is rated on a 5-point Likert scale, and the total score is converted to a 0–100 range. Higher scores indicate greater usability. SUS serves as the primary outcome measure to evaluate system usability, with a threshold of 70 considered ‘acceptable’ and ≥80 indicating ‘excellent’ usability in healthcare technology contexts.

#### 5.4.3. Secondary Outcomes

Secondary outcomes include:**User Experience Questionnaire–Short (UEQ-S):** Evaluates emotional satisfaction and user experience [100] across pragmatic quality (efficiency, perspicuity) and hedonic quality (novelty, stimulation) dimensions, assessed at T1.**Mini-Mental State Examination (MMSE):** General cognitive function [101]. Scores range from 0–30, with baseline eligibility restricted to 18–25 to capture MCI population. Potential practice effects over the 6-week interval will be acknowledged in interpretation, with primary emphasis on usability outcomes.**Trail Making Test (TMT-A/B):** Visual attention and executive function [102]. Shorter completion times indicate better performance. TMT-B/TMT-A ratio provides executive function index.**Digit Span Test (forward/backward):** Short-term and working memory [103]. Forward and backward scores are analyzed separately to distinguish memory types.**Positive and Negative Affect Schedule (PANAS):** Positive and negative affective states [104]. Higher scores reflect higher frequency of corresponding emotions. Used to assess emotional changes following intervention, with alert thresholds set at negative affect increases ≥ 1.5 SD above baseline.**Basic Psychological Needs Satisfaction Scale (BPNS):** The 21-item BPNS assesses satisfaction of three psychological needs central to Self-Determination Theory: autonomy (7 items), competence (6 items), and relatedness (8 items) [105,106]. Each item is rated on a 5-point Likert scale, with higher scores indicating greater need satisfaction. The scale explores whether participants report need satisfaction during system use, providing preliminary evidence of theoretical alignment between design features and SDT constructs, administered at T1.

#### 5.4.4. Exploratory Digital Adherence Metrics

Engagement metrics include the following:**Session completion rate:** Percentage of scheduled sessions (30 total) completed.**Mean session duration:** Average time spent per session (target: 25–35 min).**Task completion rate:** Percentage of initiated cognitive tasks completed without premature termination.**Agent interaction frequency:** Number of user-initiated interactions with each agent type (Teacher, Coach, Companion).**Response latency trends:** Changes in user response times across the intervention period, indicating cognitive adaptation or fatigue.**Digital performance indicators:** The system automatically logs task-level accuracy, response time, and improvement trajectories across cognitive exercises (episodic memory, working memory, attention, and language tasks specified in Section 5.3). These digitally collected metrics, generated through real-time user interaction tracking, will be analyzed alongside neuropsychological assessments (MMSE, TMT, Digit Span) to evaluate cognitive intervention effects and identify performance patterns not captured by traditional testing.

Adherence correlations with outcomes will be explored.

#### 5.4.5. Assessment Schedule

**T0 (Baseline):** MMSE, TMT-A/B, Digit Span, PANAS, demographic questionnaire, digital literacy assessment (MDPQ-16), technology anxiety screening.**Intervention Phase (Weeks 1–6):** Continuous digital log collection; weekly check-in calls to monitor adverse events, cognitive fatigue, emotional state (PANAS), and technology issues.**T1 (Post-intervention, Week 6):** MMSE, TMT-A/B, Digit Span, PANAS, SUS, UEQ-S, semi-structured exit interview.

### 5.5. Data Analysis Plan

Primary analysis will employ descriptive statistics for feasibility and usability outcomes (SUS mean, 95% CI; adherence rates). Secondary analyses will explore cognitive and emotional changes using paired *t*-tests if normality assumptions are satisfied (Shapiro–Wilk test, p>0.05); otherwise, Wilcoxon signed-rank tests will be used.

Analyses will include all enrolled participants (modified intention-to-treat approach), with missing data addressed through multiple imputation if the missing rate < 30%; otherwise, complete case analysis with sensitivity testing is performed. Feasibility criteria: (1) session completion ≥ 70%, (2) SUS ≥ 70, and (3) no sustained negative affect requiring discontinuation.

Exit interviews are thematically analyzed for usability barriers and system refinement suggestions.

### 5.6. Implementation Safeguards: Ethical and Safety Protocols

These protocols specify implementation safeguards for AI-based cognitive interventions in older adults: digital literacy assessment, emotional safety monitoring, and data privacy protections for vulnerable populations.

#### 5.6.1. Informed Consent Procedures

Institutional Review Board (IRB) approval is required for studies implementing this protocol, with adherence to the Declaration of Helsinki principles. The protocol specifies that participants will receive explanations of the study purpose, procedures, risks, and benefits before providing informed consent. The consent process will address AI system capabilities, limitations, and risks (including hallucinations).

#### 5.6.2. Data Privacy and Protection

The protocol specifies that all personal data will be anonymized using unique participant identifiers and handled exclusively for research purposes. Conversational logs and behavioral data will be encrypted (AES-256 standard) [107] during transmission and storage, with role-based access control limiting data access to authorized research personnel only. Participant data will not be used for model training or algorithmic adaptation without explicit additional consent.

#### 5.6.3. Safety Monitoring and Discontinuation

Adverse event monitoring includes weekly check-in calls and participant-initiated contact protocols. Discontinuation criteria include (1) participant request, (2) sustained high fatigue (self-reported score ≥ 7/10 on visual analog scale for three consecutive sessions), (3) negative affect elevation ≥ 1.5 SD above baseline on two consecutive weekly assessments, or (4) any technology-related distress judged by investigators to compromise participant well-being.

**Multi-Agent Coordination Safeguards.** To address risks inherent in multi-agent systems, the following protocols are implemented:

*Response Conflict Detection:* Cross-agent validation ensures consistency. When agents generate potentially contradictory guidance (e.g., Coach suggesting activity while Teacher indicates fatigue), the system prioritizes safety-conservative responses and alerts research staff.

*Hallucination Monitoring:* All AI-generated responses undergo automated safety filtering in real-time. Flagged responses (containing medical advice, financial guidance, or factually implausible content) are blocked from delivery, with a safe fallback message displayed to the user (e.g., “I’m not certain about that. Please consult with your healthcare provider.”). Blocked responses trigger immediate notification to research staff for review. Research staff conduct weekly audits of interaction logs to identify systemic issues.

*Response Fidelity Monitoring:* To ensure consistent agent behavior, research staff conduct weekly random sampling of 10% of interaction logs, assessing (1) persona consistency (whether Coach, Teacher, Companion responses align with their specified roles), (2) response appropriateness (whether feedback matches user performance and emotional state), and (3) adherence to conversational design guidelines (Section 3.3.3). Deviations from specified behavior trigger system review and potential parameter adjustment.

*User Confusion Management:* Weekly assessments include agent recognition questions (“Which agent helps with daily activities?”). If participants demonstrate confusion or express a preference for fewer agents, the system supports fallback to single-agent mode (typically Companion for emotional support).

*Escalation Workflow:* Research staff receive automated alerts for (1) repeated task failures suggesting system malfunction, (2) participant expressions of distress or frustration, and (3) technical errors in agent coordination. Staff contact participants within 24 h of alert generation.

#### 5.6.4. Data Sharing and Dissemination

De-identified datasets and system usability logs will be made available to qualified researchers upon reasonable request via the corresponding author, subject to compliance with privacy protection laws and institutional data governance frameworks. System implementation code and protocol documentation are publicly available (see Section 3.2.3).

Study outcomes will be reported following CONSORT-AI and SPIRIT-AI guidelines and disseminated through peer-reviewed publications and scientific conferences.

## 6. Discussion

### 6.1. Principal Findings

This protocol proposes addressing implementation challenges in digital cognitive interventions for older adults. The CTC Framework is designed to integrate behavioral coaching, adaptive cognitive training, and emotional support through a modular architecture intended to enable investigation of how these components may contribute to sustained engagement.

The protocol specifies implementation safeguards for vulnerable populations: digital literacy assessment (MDPQ-16), technology anxiety monitoring (CARS), emotional safety protocols with discontinuation criteria, adaptive difficulty targeting an 85% success rate, and data privacy protections.

### 6.2. Implementation Safeguards for AI-Based Interventions

This protocol proposes systematic safeguards for implementing AI-based cognitive interventions in older adults, addressing four key challenges: digital literacy barriers, emotional safety risks, cognitive load management, and data privacy protections.

#### 6.2.1. Digital Literacy and Technology Anxiety Assessment

Baseline digital literacy assessment enables valid usability evaluation by controlling for prior device experience, which could otherwise confound primary outcomes. The MDPQ-16 provides standardized measurement enabling stratified analysis [94]. Technology anxiety screening (CARS) identifies participants requiring additional support, with elevated anxiety serving as a discontinuation criterion [5].

#### 6.2.2. Emotional Safety Monitoring in AI-Driven Interventions

Systematic emotional monitoring is necessary given the Companion’s capacity to influence affective states. Weekly PANAS assessment with alert thresholds (negative affect ≥ 1.5 SD above baseline) enables early detection of adverse emotional responses [104].

#### 6.2.3. Cognitive Load Management and Adaptive Difficulty

The 85% success rate target, based on optimal challenge principles, aims to maintain engagement while preventing frustration [67]. Cognitive load monitoring through response latency, error rates, and attention variance is designed to enable dynamic difficulty adjustment. The adaptive difficulty algorithm (Equation (Equation 2)) adjusts task complexity based on individual performance.

#### 6.2.4. Data Privacy and Ethical AI Deployment

Vulnerable populations require data protection beyond standard research protocols. This protocol restricts using participant data for model training without explicit consent [21].

Informed consent procedures will address AI system limitations, including hallucination risks. All conversational and behavioral data will be encrypted (AES-256), anonymized using unique identifiers, and accessible only to authorized research personnel.

### 6.3. Limitations

This protocol has several limitations. First, the six-week single-arm design prioritizes feasibility assessment over causal inference. Observed changes cannot be attributed to the intervention without a control group, requiring future RCTs with active control conditions.

Second, the framework requires sustained internet connectivity (approximately 3–5 Mbps) and baseline device proficiency, excluding older adults without these resources. This study targets barrier reduction within a subpopulation with minimal digital access. Cloud API operational costs for the prototype are estimated at $0.15–$0.30 per 30 min session, substantially lower than therapist-delivered cognitive training (approximately $50 per session in South Korea), enabling potential implementation through public infrastructure such as community health centers. However, populations in complete offline isolation remain outside the intervention scope.

Third, system parameters (cognitive load weights, adaptive difficulty thresholds, agent configurations) represent theory-informed proposals requiring empirical validation. The feasibility study employs fixed parameters to assess usability and safety, with parameter optimization planned for subsequent efficacy trials based on observed interaction patterns. Multi-agent coordination at scale may pose technical challenges, including response latency and inconsistencies.

Fourth, interface accessibility specifications are guided by WCAG 2.1 AA standards but have not undergone a formal accessibility audit or certification. The feasibility study will assess practical usability with older adults, with a comprehensive accessibility evaluation planned for subsequent implementation phases.

### 6.4. Protocol Adaptability and Future Validation

This protocol represents the feasibility stage of the CTC Framework, including a six-week pilot phase as a preliminary clinical implementation. It supports a stepwise validation pathway leading to subsequent large-scale clinical trials. This progression aligns with methodological recommendations for digital therapeutics.

Future trials should compare the framework with conventional cognitive training or single-agent alternatives, assessing cognitive outcomes, functional outcomes, and quality of life. The protocol supports adaptation to different contexts. Investigators may adjust intervention duration, session frequency, or agent configuration based on research questions while maintaining core architecture and safety protocols.

### 6.5. Conclusions

This protocol proposes technical architecture and implementation procedures for AI-based cognitive interventions in aging populations with baseline digital access. The CTC Framework aims to integrate accessibility-focused design, adaptive difficulty, and multi-agent support to explore approaches to reducing interaction barriers, overcoming static training limitations, and enabling personalized adaptation in digital cognitive interventions.

The protocol specifies implementation safeguards for digital literacy assessment, technology anxiety management, emotional safety monitoring, and data privacy protections, intended to provide clinicians and researchers with methodological guidance for safely implementing AI-based interventions. Feasibility outcomes will inform subsequent RCT design through usability metrics, adherence patterns, and variance estimates for power calculations.

No empirical outcomes are available at this protocol stage. The framework’s effectiveness requires validation through the planned feasibility study and future controlled trials.

## Figures and Tables

**Figure 1 healthcare-13-03225-f001:**
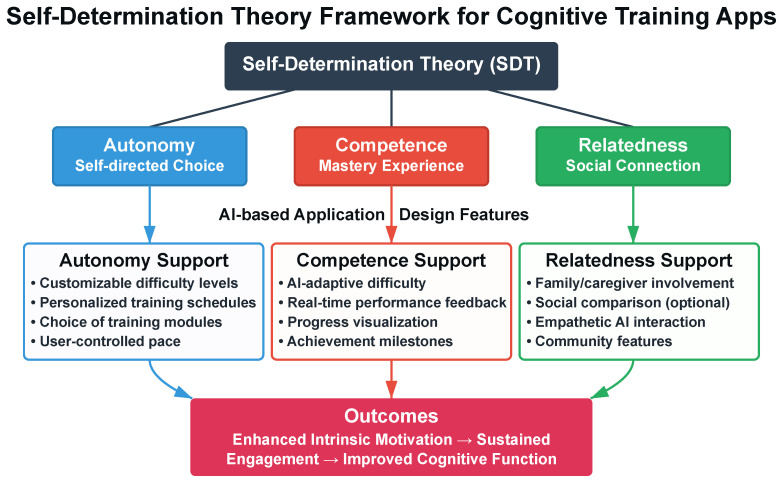
Theoretical framework linking Self-Determination Theory to AI-based cognitive training design. The three fundamental psychological needs (autonomy, competence, and relatedness) are proposed to be addressed through specific application design features, theoretically contributing to intrinsic motivation and sustained engagement. The CTC Framework adopts this theoretical model, with empirical validation planned through the feasibility study.

**Figure 2 healthcare-13-03225-f002:**
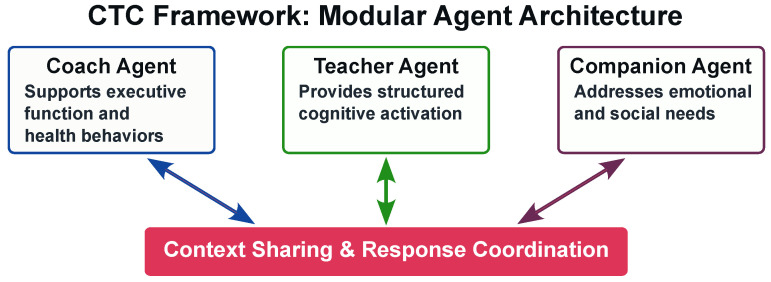
CTC Framework: Modular agent architecture with functional specialization. The three agents (Coach, Teacher, and Companion) are designed to share contextual information through a central coordination layer, intended to enable adaptive and coordinated responses based on user state.

**Figure 3 healthcare-13-03225-f003:**
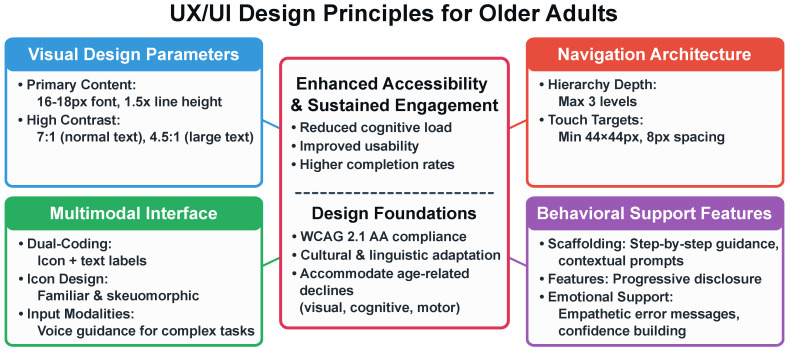
UX/UI design principles for older adults across four domains: Visual Design (text sizing, contrast ratios), Navigation Architecture (hierarchy depth, touch targets), Multimodal Interface (dual-coding, icon design, voice guidance), and Behavioral Support (scaffolding, progressive disclosure, empathetic feedback). Design specifications are guided by WCAG 2.1 AA standards [83].

**Figure 4 healthcare-13-03225-f004:**
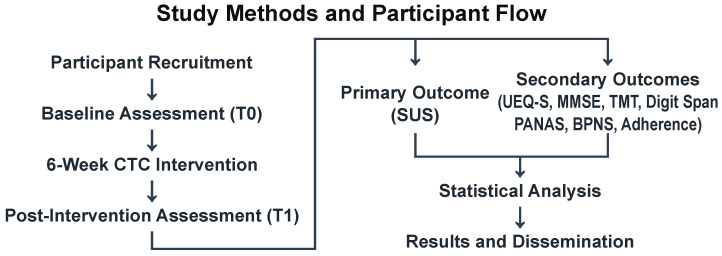
Study Protocol Flowchart. The CTC Framework feasibility evaluation includes participant screening and enrollment (*n*
=14), baseline assessment (T0) with cognitive and digital literacy measures, six-week intervention (5 sessions/week, 30 min/session), post-intervention assessment (T1), and data analysis. Primary outcome: System Usability Scale (SUS). Secondary outcomes: user experience, cognitive function, emotional state, needs satisfaction, and adherence metrics. Approximately 10% attrition anticipated.

**Table 1 healthcare-13-03225-t001:** CTC Framework: Three agents (personas) with primary functions and implementation approaches.

Agent (Persona)	Primary Functions	Implementation Approach	Example Interaction
Coach	Daily activity management, behavioral nudges, adherence support	Supports executive function through implementation intentions	“Let’s plan tomorrow’s schedule together”
Teacher	Cognitive training: memory, attention, language	Adaptive difficulty with 85% success target	“Recall three items from the morning list”
Companion	Emotional support, reminiscence, social engagement	Empathetic interaction with affect monitoring	“Tell me about your favorite memory”

## Data Availability

Implementation code for the context-adaptive cognitive flow system is publicly available at https://github.com/jeongtaeksoo/context-adaptive-cognitive-flow (accessed on 7 December 2025). Future studies will make de-identified data available upon reasonable request as described in Section 5.6.4.

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
