# Peer review of "A Generative AI Framework for Cognitive Intervention in Older Adults: An Integrated Engineering Design and Clinical Protocol"

_healthcare, 2025, doi:10.3390/healthcare13243225_

Round 1

Reviewer 1 Report

Comments and Suggestions for Authors

Dear Authors, Thank you for the opportunity to review this protocol manuscript. The topic-AI-driven cognitive intervention for older adults-is timely, socially important, and highly relevant to digital health and aging research. The proposed Coach–Teacher–Companion (CTC) Framework presents a compelling multi-agent generative AI structure, and the manuscript demonstrates strong integration of cognitive science, human–computer interaction, and clinical protocol design.Overall, the manuscript has significant merit and originality. Below are specific suggestions to strengthen clarity, scientific rigor, and methodological transparency.

1. Introduction

The introduction provides extensive background and cites relevant recent studies. However, it is longer than necessary for a protocol paper and sometimes reads like a full systematic review. Consider streamlining by:

  • reducing repeated discussion of digital exclusion mechanisms,

  • shortening the historical/contextual overview,

  • emphasizing the specific gap the CTC Framework fills (e.g., multi-agent adaptive system + standardized implementation protocol).

A more focused introduction would improve readability and make the narrative more concise.

2. Research Design

The overall research design—a 6-week single-arm feasibility study—is appropriate for early-stage AI interventions in older adults. The study flowchart and overall structure are clear.

However, please consider clarifying:

  • why the specific intervention duration (6 weeks) and frequency (5 sessions/week) were chosen,

  • whether this schedule aligns with cognitive intervention literature or pilot study conventions,

  • how expected feasibility outcomes (e.g., adherence thresholds, usability benchmarks) were derived.

Providing citations or conceptual justification will strengthen the rationale.

3. Methods

The Methods section is detailed, but several areas would benefit from additional clarification:

(a) Sample size justification

While feasibility studies often use small samples, adding quantitative context (e.g., expected SUS variability, confidence interval width, or pilot guidelines such as Julious 2005 “12 per group” rule) would strengthen the justification for n=14.

(b) Technology anxiety assessment

Digital literacy assessment with MDPQ-16 is excellent. However, the manuscript references “technology anxiety” without specifying a validated scale. Introducing a clear, standardized instrument would enhance reproducibility.

(c) Cognitive assessment considerations

MMSE may exhibit practice effects over a 6-week interval. Consider:

  • including alternative forms, or

  • discussing how potential practice effects will be interpreted.

(d) AI system fidelity & hallucination safeguards

Given this is an AI-based intervention, readers would benefit from:

  • a fidelity monitoring plan to ensure consistent agent responses,

  • a protocol for handling hallucinations, inappropriate outputs, or safety-related content,

  • clarification on how the research team will audit or review AI-generated interactions.

These points are essential for clinical reliability and safety.

(e) Description of intervention content

The paper explains the roles of the Coach/Teacher/Companion agents in high-level terms. For replication, consider adding:

  • sample prompts or exercise types,

  • examples of daily activity–embedded cognitive tasks,

  • examples of reminiscence or emotional support dialogue.

This will make the intervention more concrete for future researchers.

4. Figures and Tables

Figures are generally clear and well-designed. Some diagrams (particularly the multi-agent architecture) could be simplified to reduce visual complexity. Consider ensuring:

  • consistent formatting across figures,

  • slightly larger font sizes for readability.

  • 5. Conclusions

    The conclusions are appropriate for a protocol paper. You may wish to emphasize:

    -how the feasibility results will inform the design of the subsequent RCT,

    -what specific refinements you expect to evaluate based on pilot data (e.g., engagement patterns, difficulty adaptation, emotional safety).
  • Major Concerns & Areas Needing Revision

    1. The Paper Is Very Long for a Protocol Article

    The manuscript exceeds typical protocol paper length (24 pages before references).
    Sections such as the theoretical framework and UI/UX design could be shortened or moved to supplementary material.

    2. Lack of Empirical Evidence for Some Design Decisions

    Examples:

    • Cognitive load equation weights (w₁, w₂, w₃)

    • Threshold of 85% success as universal target

    • Persona count fixed at three

    • Valence-arousal thresholds
      These are reasonable choices, but require stronger justification or clarification that they are hypothetical and require empirical validation.

    3. Limited Discussion of Risks Unique to Multi-Agent AI

    Potential issues not discussed:

    • contradictory messages between agents

    • user confusion with three personas

    • cognitive overload for frail/MCI patients

    • hallucination management and fallback workflows

    You mention these risks but do not provide mitigation strategies.

    4. Sample Size Rationale Needs Strengthening

    You cite guidelines for feasibility studies, but reviewers may request:

    • references supporting “≈12 participants” rule

    • justification that 14 participants can reliably estimate adherence/usability

    • explanation of how variance estimates will inform future RCT sample sizes

    5. Overuse of Prior Literature in Early Sections

    The Background section reads like a full literature review rather than a protocol introduction. It may be improved by being more concise.

    6. Missing Detail on Actual AI Models

    The manuscript references:

    • Whisper for ASR

    • Distilled language model

    • Persona-specific adaptations

    • Multimodal sensing

    But it does not specify:

    • base LLM model used

    • whether the system runs on-device or cloud-based

    • inference cost and feasibility for deployment
      This detail is important for reproducibility.

    7. No Clear Description of Intervention Content Library

    For example:

    • what cognitive tasks are used?

    • what types of daily-activity–embedded exercises?

    • examples of reminiscence interaction prompts?

    Providing a few structured examples would greatly clarify the intervention.

    Minor Comments

    Clarity & Writing

    • Several paragraphs are dense; consider trimming jargon.

    • Ensure consistent use of “older adults”, “seniors”, “aging population”.

    • The Conclusion could be more concise.

    Figures & Tables

    • Figure 1 and Figure 3 are effective, but Figure 2 needs clearer labels.

    • Table 1 could be expanded to include examples or use cases.

    Ethics

    • Clarify how hallucinations are handled in real time (e.g., refusal, escalation, safe responses).

    Feasibility Study Procedures

    • Weekly PANAS monitoring is appropriate, but specify administrator (phone vs. in-app).

    • Clarify whether sessions are supervised or unsupervised.

Comments on the Quality of English Language

The manuscript is written well overall, but several sections-particularly the theoretical framework-could be made more concise. Simplifying long technical paragraphs will improve clarity and accessibility. 

The English language throughout the manuscript is generally clear and understandable, and the technical terminology is used appropriately. However, several sections-especially the Introduction and Background-are overly lengthy and contain complex or repetitive sentence structures that make the text difficult to follow at times. Streamlining the wording, improving paragraph flow, and reducing redundancy would enhance readability and clarity. While there are no major grammatical issues, careful editing for conciseness and smoother expression is recommended.

Author Response

AUTHOR'S NOTES TO REVIEWER 1

Dear Reviewer 1,

Thank you for your thorough and constructive review of our protocol manuscript. We greatly appreciate your recognition of the manuscript's merit and originality, as well as your detailed suggestions for strengthening clarity, scientific rigor, and methodological transparency. We have carefully addressed each of your comments and revised the manuscript accordingly. Below, we provide point-by-point responses.

---

GENERAL COMMENTS

Comment 1: Introduction
"The introduction provides extensive background and cites relevant recent studies. However, it is longer than necessary for a protocol paper and sometimes reads like a full systematic review. Consider streamlining by: reducing repeated discussion of digital exclusion mechanisms, shortening the historical/contextual overview, emphasizing the specific gap the CTC Framework fills (e.g., multi-agent adaptive system + standardized implementation protocol). A more focused introduction would improve readability and make the narrative more concise."

Response 1:
Thank you for the suggestion. We have streamlined the introduction to 5 concise paragraphs by reducing repeated discussion of digital exclusion mechanisms, removing historical/contextual overview, and emphasizing the specific gap the CTC Framework fills. The revised introduction now clearly states the protocol's objectives in the final paragraph (Section 1):

"The protocol aims to establish implementation methodology for AI-based cognitive interventions in older adults. Specific objectives include: specifying participant screening procedures (digital literacy assessment, technology anxiety monitoring), defining safety protocols (cognitive load thresholds, discontinuation criteria), establishing data privacy safeguards, and designing a feasibility study to evaluate usability and preliminary outcomes."

---

Comment 2: Research Design
"The overall research design—a 6-week single-arm feasibility study—is appropriate for early-stage AI interventions in older adults. The study flowchart and overall structure are clear. However, please consider clarifying: why the specific intervention duration (6 weeks) and frequency (5 sessions/week) were chosen, whether this schedule aligns with cognitive intervention literature or pilot study conventions, how expected feasibility outcomes (e.g., adherence thresholds, usability benchmarks) were derived. Providing citations or conceptual justification will strengthen the rationale."

Response 2:
Thank you for the suggestion. We have added justification for intervention duration and frequency in Section 5.3:

"Participants complete 30 sessions over six weeks (five sessions/week, 30 minutes/session). This schedule is designed to generate sufficient interaction data (15 total hours) for feasibility assessment while minimizing participant burden, consistent with pilot study guidelines (Bowen et al., 2009; Lampit et al., 2014)."

We have clarified feasibility outcome thresholds in Section 5.1:

"The primary objective is to assess feasibility through: (1) adherence (≥70% session completion), (2) usability (SUS ≥70), and (3) emotional safety (negative affect within predefined thresholds)."

---

METHODS - SPECIFIC AREAS

Comment 3a: Sample size justification
"While feasibility studies often use small samples, adding quantitative context (e.g., expected SUS variability, confidence interval width, or pilot guidelines such as Julious 2005 '12 per group' rule) would strengthen the justification for n=14."

Response 3a:
Thank you for the suggestion. We have strengthened the sample size justification in Section 5.2.3:

"Following established recommendations for single-arm pilot studies, approximately 12 participants are needed to achieve adequate precision in estimating variance and retention rates. Given the single-arm design and practical constraints, a total of 14 participants will be enrolled, allowing for anticipated 10% attrition while maintaining adequate sample size for feasibility assessment. This sample size is consistent with pilot-study recommendations (Kunselman et al., 2024) and provides information to refine study procedures before progressing to a larger RCT."

---

Comment 3b: Technology anxiety assessment
"Digital literacy assessment with MDPQ-16 is excellent. However, the manuscript references 'technology anxiety' without specifying a validated scale. Introducing a clear, standardized instrument would enhance reproducibility."

Response 3b:
Thank you for the suggestion. We have specified the Computer Anxiety Rating Scale (CARS) in Section 5.2.2:

"Technology anxiety is measured at baseline with the Computer Anxiety Rating Scale (CARS) (Heinssen et al., 1987), with scores >2 SD above normative means triggering enhanced onboarding support. For participants meeting this threshold, weekly CARS assessments are administered during check-in calls to monitor anxiety trajectories and identify need for additional support or discontinuation."

---

Comment 3c: Cognitive assessment considerations
"MMSE may exhibit practice effects over a 6-week interval. Consider: including alternative forms, or discussing how potential practice effects will be interpreted."

Response 3c:
Thank you for the suggestion. We have acknowledged this in Section 5.4.3:

"Mini-Mental State Examination (MMSE): General cognitive function (Folstein et al., 1975). Scores range from 0–30, with baseline eligibility restricted to 18–25 to capture MCI population. Potential practice effects over the 6-week interval will be acknowledged in interpretation, with primary emphasis on usability outcomes."

---

Comment 3d: AI system fidelity & hallucination safeguards
"Given this is an AI-based intervention, readers would benefit from: a fidelity monitoring plan to ensure consistent agent responses, a protocol for handling hallucinations, inappropriate outputs, or safety-related content, clarification on how the research team will audit or review AI-generated interactions. These points are essential for clinical reliability and safety."

Response 3d:
Thank you for the feedback. We have expanded Section 5.6.3 (Safety Monitoring and Discontinuation) to include multi-agent coordination safeguards:

1. Response Conflict Detection: Cross-agent validation ensures consistency, with safety-conservative responses prioritized when agents generate potentially contradictory guidance.

2. Hallucination Monitoring: "All AI-generated responses undergo automated safety filtering in real-time. Flagged responses (containing medical advice, financial guidance, or factually implausible content) are blocked from delivery, with a safe fallback message displayed to the user (e.g., 'I'm not certain about that. Please consult with your healthcare provider.'). Blocked responses trigger immediate notification to research staff for review. Research staff conduct weekly audits of interaction logs to identify systemic issues."

3. Response Fidelity Monitoring: "To ensure consistent agent behavior, research staff conduct weekly random sampling of 10% of interaction logs, assessing: (1) persona consistency (whether Coach, Teacher, Companion responses align with their specified roles), (2) response appropriateness (whether feedback matches user performance and emotional state), and (3) adherence to conversational design guidelines (Section 3.3.3). Deviations from specified behavior trigger system review and potential parameter adjustment."

4. User Confusion Management: Weekly assessments include agent recognition questions, with fallback to single-agent mode if participants demonstrate confusion.

5. Escalation Workflow: Automated alerts for system malfunction, participant distress, or technical errors, with staff contact within 24 hours.

---

Comment 3e: Description of intervention content
"The paper explains the roles of the Coach/Teacher/Companion agents in high-level terms. For replication, consider adding: sample prompts or exercise types, examples of daily activity–embedded cognitive tasks, examples of reminiscence or emotional support dialogue. This will make the intervention more concrete for future researchers."

Response 3e:
Thank you for the suggestion. We have added intervention content examples:

1. Table 1 (Section 3.2.1) now includes an "Example Interaction" column for each agent:
- Coach: "Let's plan tomorrow's schedule together"
- Teacher: "Recall three items from the morning list"
- Companion: "Tell me about your favorite memory"

2. Section 5.3 now includes cognitive exercise descriptions:
"Cognitive exercises include: (1) episodic memory tasks (recalling grocery lists, medication schedules), (2) working memory tasks (backward digit span embedded in phone number entry), (3) attention tasks (identifying target items during simulated shopping), and (4) language tasks (word generation during recipe planning). Task difficulty adapts based on performance as specified in Section 3.2.2."

---

MAJOR CONCERNS

Comment 4: The Paper Is Very Long for a Protocol Article
"The manuscript exceeds typical protocol paper length (24 pages before references). Sections such as the theoretical framework and UI/UX design could be shortened or moved to supplementary material."

Response 4:
Thank you for the suggestion. We have condensed the manuscript by streamlining the Introduction (Section 1), condensing the Background section (Section 2), tightening the System Architecture section (Section 3), and merging redundant paragraphs throughout. The manuscript remains detailed due to presenting both technical architecture and clinical protocol, but it is now 16 pages before references (entire pages reduced: 24->22).

---

Comment 5: Lack of Empirical Evidence for Some Design Decisions
"Examples: Cognitive load equation weights (w₁, w₂, w₃), Threshold of 85% success as universal target, Persona count fixed at three, Valence-arousal thresholds. These are reasonable choices, but require stronger justification or clarification that they are hypothetical and require empirical validation."

Response 5:
Thank you for the suggestion. We have acknowledged that these parameters are theory-derived proposals requiring validation:

1. Cognitive load weights (Section 3.2.2): "Initial weight parameters (w₁ = 0.4, w₂ = 0.35, w₃ = 0.25) prioritize processing speed as the primary indicator, consistent with its established role as the primary mediator of age-related cognitive performance (Salthouse, 1996). The feasibility study will assess whether this theoretically-derived weighting scheme appropriately captures cognitive load variations and identify necessary refinements for subsequent efficacy trials."

2. 85% success target (Section 3.2.2): "The system is designed to target success probability P* at approximately 0.85, based on optimal challenge principles (Wilson et al., 2019). Individual calibration during the feasibility study may reveal need for adjustment based on participant characteristics and task domains."

3. Three-agent configuration (Section 3.2.1): "The optimal agent configuration for this population requires empirical validation through the feasibility study."

4. Design parameters (Section 6.3): "Design parameters (cognitive load weights, adaptive difficulty thresholds, agent configurations) are theory-derived proposals requiring empirical validation."

---

Comment 6: Limited Discussion of Risks Unique to Multi-Agent AI
"Potential issues not discussed: contradictory messages between agents, user confusion with three personas, cognitive overload for frail/MCI patients, hallucination management and fallback workflows. You mention these risks but do not provide mitigation strategies."

Response 6:
Thank you for the feedback. We have addressed this in Section 5.6.3 (Multi-Agent Coordination Safeguards), as detailed in Response 3d above, including response conflict detection, hallucination monitoring, response fidelity monitoring, user confusion management, and escalation workflow.

---

Comment 7: Sample Size Rationale Needs Strengthening
"You cite guidelines for feasibility studies, but reviewers may request: references supporting '≈12 participants' rule, justification that 14 participants can reliably estimate adherence/usability, explanation of how variance estimates will inform future RCT sample sizes."

Response 7:
Thank you for the suggestion. We have addressed this in Section 5.2.3 as detailed in Response 3a, citing the "approximately 12 participants" guideline (Kunselman et al., 2024). Section 6.5 now states: "Feasibility outcomes will inform subsequent RCT design through usability metrics, adherence patterns, and variance estimates for power calculations."

---

Comment 8: Overuse of Prior Literature in Early Sections
"The Background section reads like a full literature review rather than a protocol introduction. It may be improved by being more concise."

Response 8:
Thank you for the suggestion. We have condensed Section 2 by merging redundant paragraphs, removing excessive citations, consolidating discussion of generative AI capabilities, and streamlining the Self-Determination Theory section.

---

Comment 9: Missing Detail on Actual AI Models
"The manuscript references: Whisper for ASR, Distilled language model, Persona-specific adaptations, Multimodal sensing. But it does not specify: base LLM model used, whether the system runs on-device or cloud-based, inference cost and feasibility for deployment. This detail is important for reproducibility."

Response 9:
Thank you for the suggestion. We have added technical specifications in Section 3.2.3:

"The Response Engine employs cloud-based Application Programming Interface (API) access to LLMs (GPT-4o-mini, Gemini 1.5 Flash, or equivalent models) without fine-tuning. Each agent (Coach, Teacher, Companion) uses persona-specific prefix modules requiring <0.1% additional parameters (Han et al., 2023). The API-based architecture ensures user data is not retained for model training, consistent with data privacy requirements. Estimated operational cost is \$0.15–\$0.30 per 30-minute session. Response generation incorporates safety constraints for age-appropriate content (Ngo et al., 2021). Implementation code is available at [repository citation]."

---

Comment 10: No Clear Description of Intervention Content Library
"For example: what cognitive tasks are used? what types of daily-activity–embedded exercises? examples of reminiscence interaction prompts? Providing a few structured examples would greatly clarify the intervention."

Response 10:
Thank you for the suggestion. This has been addressed through the additions described in Response 3e.

---

MINOR COMMENTS

Comment 11: Figures & Tables
"Figure 1 and Figure 3 are effective, but Figure 2 needs clearer labels. Table 1 could be expanded to include examples or use cases."

Response 11:
Thank you for the suggestion. We have addressed these points:
- Figure 2 caption has been enhanced to clarify the context sharing mechanism
- Table 1 now includes an "Example Interaction" column
- All figures (Figures 1–4) have been revised with improved visual clarity and larger font sizes

---

Comment 12: Ethics - Hallucinations
"Clarify how hallucinations are handled in real time (e.g., refusal, escalation, safe responses)."

Response 12:
Thank you for the suggestion. This has been addressed in Section 5.6.3 as detailed in Response 3d.

---

Comment 13: Feasibility Study Procedures
"Weekly PANAS monitoring is appropriate, but specify administrator (phone vs. in-app). Clarify whether sessions are supervised or unsupervised."

Response 13:
Thank you for the suggestion. We have clarified:

1. Section 5.4.5: "Intervention Phase (Weeks 1–6): Continuous digital log collection; weekly check-in calls to monitor adverse events, cognitive fatigue, emotional state (PANAS), and technology issues." (PANAS administered during phone calls)

2. Section 5.3: "Following onboarding, participants complete all 30 intervention sessions independently using their personal devices at home, with research staff available via phone for technical support."

---

We thank you again for your thorough review. If you have any further questions, we would be happy to address them.

Respectfully,
Taeksoo Jeong, Geonhwi Hwang and Doo Young Kim

Reviewer 2 Report

Comments and Suggestions for Authors

I would like to congratulate the authors for submitting a manuscript addressing an emerging topic in AI-supported cognitive interventions for older adults. The comments below are intended to offer constructive, detailed feedback on specific lines and sections of the manuscript:

ABSTRACT (Lines 1–27)

  1. Lines 5–9: The objective section implies completed outcomes (“presents the CTC Framework”), but this is a protocol. The authors should replace with “This protocol aims to…”
  2. Lines 18–25: The conclusions imply demonstrated effectiveness (“integrates accessibility design… addresses digital exclusion”). These should be reframed as intended aims, not proven results.
  3. The authors should add one sentence clarifying that no results are available yet and this is a protocol-only manuscript.

 INTRODUCTION (Lines 28–85)

  1. Lines 31–36: The statistical results are strong, but the introduction jumps too quickly to dropout rates (Lines 32–33) without fully establishing why digital tools matter clinically.
  2. Lines 37–47: Very detailed discussion of barriers; The authors should consider reducing detail here and expanding in a “Background” or “Rationale” subsection.
  3. Lines 49–59: Several rhetorical questions are posed; in scientific writing, these should be converted into a single, clear knowledge gap.
  4. Lines 78–85: The aims of the protocol are not explicitly stated. The authors should add a clear paragraph ending the introduction with:
  • overall aim of the protocol
  • specific objectives
  • expected contribution.

 BACKGROUND & THEORETICAL FRAMEWORK (Lines 86–186)

  1. Lines 127–140: Ethical risks of hallucination are mentioned only briefly. The authors should expand, especially given the target population.
  2. Lines 154–186: This section reads like Methods rather than Background; The authors should consider relocating detailed protocol components to Section 3.

 SYSTEM ARCHITECTURE (Lines 187–371)

  1. Lines 188–199: “Design Objectives” are strong, but the section includes methodological claims (usability, ecological validity) without data. Please clarify these are design assumptions.
  2. Lines 226–236: Emotional support mechanisms are described convincingly; though, please state that the efficacy of these mechanisms remains unvalidated until feasibility testing.
  3. Lines 283–295 (Table 1): Table 1 is clear, but lacks:
    • example prompts,
    • which AI model each agent uses (Also not included:
      – what version
      – model size
      – local vs. API-based
      – any fine-tuning or adaptation),
    • decision-boundary logic for agent switching.

INTERFACE & INTERACTION DESIGN (Lines 372–415)

  1. Lines 402–410: Conversational style guidelines are appropriate, but should clarify that these principles are design intentions, not validated outcomes.

PROTOTYPE IMPLEMENTATION (Lines 416–449)

  1. Lines 417–425: Good conceptual walkthrough, but some descriptions (e.g., “Teacher adjusts difficulty based on emotional assessment”) could be misread as empirical results.
  2. Lines 433–444: Simulation scenarios should be clearly labeled as demonstrations, not observations.

METHODS (Lines 450–596)

  1. Lines 455–467: The study design is appropriate, but you need an explicit statement acknowledging the limitations of a single-arm feasibility design.
  2. Lines 468–495: Inclusion criteria are clear; however, the requirement for device ability may systematically exclude the very population most at risk (digitally excluded adults). The authors should acknowledge this limitation.
  3. Lines 496–505: Onboarding procedures is mentioned but not detailed; reproducibility requires a standardized script or manual.
  4. Lines 556–565: The statistical plan is underdeveloped. The authors should specify:
    • parametric vs. nonparametric tests,
    • handling of missing data,
    • rationale for using t-tests in small samples.

DISCUSSION (Lines 598–664)

  1. Lines 599–611: Good synthesis, but the paragraph reads like confirmed findings. The authors should modify language to avoid implying completed evaluation.
  2. Lines 612–639: Discussion of emotional safety, digital literacy, and adaptive difficulty is solid, but lacks acknowledgement of design limitations (AI drift, hallucinations, cultural differences).

CONCLUSIONS (Lines 664–674)

  1. Lines 664–674: Conclusions overstate what the protocol achieves. The authors should replace all definitive verbs (“integrates,” “addresses”) with conditional language.
  2. The authors should add one sentence acknowledging that no empirical outcomes exist yet and the framework’s effectiveness will require future evaluation.
Comments on the Quality of English Language

The English is good but would benefit from minor editing for clarity and flow.

Author Response

AUTHOR'S NOTES TO REVIEWER 2

Dear Reviewer 2,

Thank you for your detailed and systematic review of our protocol manuscript. We greatly appreciate your meticulous line-by-line feedback and your constructive suggestions for improving clarity, precision, and methodological rigor. We have carefully addressed each of your comments and revised the manuscript accordingly. Below, we provide point-by-point responses.

---

ABSTRACT

Comment 1: Lines 5–9 (Objective section)
"The objective section implies completed outcomes ('presents the CTC Framework'), but this is a protocol. The authors should replace with 'This protocol aims to…'"

Response 1:
Thank you for this suggestion. We have revised the Abstract Objective to use conditional language (Abstract, Objective section):

"This protocol aims to present the CTC Framework (Coach-Teacher-Companion), a tri-agent generative AI system designed for cognitive interventions in older adults."

Also, we tried to revise using conditional language throughout the protocol to clarify these are design intentions. But one thing we need to mention: In Section 1 (Introduction) the fourth paragraph opens with "we introduce the CTC Framework" to appeal our originality, but immediately follows with "The framework is designed to address barriers..." to maintain the protocol tone.

---

Comment 2: Lines 18–25 (Conclusions section)
"The conclusions imply demonstrated effectiveness ('integrates accessibility design… addresses digital exclusion'). These should be reframed as intended aims, not proven results. The authors should add one sentence clarifying that no results are available yet and this is a protocol-only manuscript."

Response 2:
Thank you for this suggestion. We have revised the Abstract Conclusions to use conditional language and added explicit acknowledgment that empirical validation is required (Abstract, Conclusions section):

"The CTC Framework is designed to provide methodological and ethical safeguards for clinical implementation, including standardized procedures for digital literacy assessment, technology anxiety management, emotional safety monitoring, and data privacy protections. Empirical validation of the framework's feasibility and efficacy is required through future studies."

We have also revised "addresses digital exclusion" to "address barriers associated with digital exclusion" throughout the manuscript to avoid overclaiming.

---

INTRODUCTION

Comment 3: Lines 31–36
"The statistical results are strong, but the introduction jumps too quickly to dropout rates (Lines 32–33) without fully establishing why digital tools matter clinically."

Response 3:
Thank you for the suggestion. We have revised the Introduction to establish clinical relevance before discussing barriers (Section 1, first two paragraphs). The second paragraph now emphasizes the recovery potential of sustained digital engagement before introducing dropout barriers.

---

Comment 4: Lines 37–47
"Very detailed discussion of barriers; The authors should consider reducing detail here and expanding in a 'Background' or 'Rationale' subsection."

Response 4:
Thank you for the suggestion. We have condensed barrier discussion in the Introduction (Section 1, third paragraph) and moved detailed discussion to Section 2 (Background and Theoretical Framework).

---

Comment 5: Lines 49–59
"Several rhetorical questions are posed; in scientific writing, these should be converted into a single, clear knowledge gap."

Response 5:
Thank you for the suggestion. We have removed rhetorical questions and converted them into declarative statements establishing the knowledge gap (Section 1, fourth paragraph):

"However, clinical implementation requires methodological frameworks addressing digital literacy assessment, cognitive load management, emotional safety monitoring, and data privacy protections."

---

Comment 6: Lines 78–85
"The aims of the protocol are not explicitly stated. The authors should add a clear paragraph ending the introduction with: overall aim of the protocol, specific objectives, expected contribution."

Response 6:
Thank you for this suggestion. We have added explicit protocol aims in the final paragraph of Section 1:

"The protocol aims to establish implementation methodology for AI-based cognitive interventions in older adults. Specific objectives include: specifying participant screening procedures (digital literacy assessment, technology anxiety monitoring), defining safety protocols (cognitive load thresholds, discontinuation criteria), establishing data privacy safeguards, and designing a feasibility study to evaluate usability and preliminary outcomes."

---

BACKGROUND & THEORETICAL FRAMEWORK

Comment 7: Lines 127–140
"Ethical risks of hallucination are mentioned only briefly. The authors should expand, especially given the target population."

Response 7:
Thank you for the suggestion. We have substantially expanded hallucination safeguards in Section 5.6.3 (Safety Monitoring and Discontinuation), including:
- Real-time automated safety filtering of AI-generated responses
- Blocking and fallback messages for flagged content (medical advice, financial guidance, factually implausible content)
- Immediate notification to research staff
- Weekly audits of interaction logs

---

Comment 8: Lines 154–186
"This section reads like Methods rather than Background; The authors should consider relocating detailed protocol components to Section 3."

Response 8:
Thank you for the suggestion. We have moved detailed protocol components from Section 2.4 to Section 3 and Section 5, retaining only essential framework rationale in Section 2.4.

---

SYSTEM ARCHITECTURE

Comment 9: Lines 188–199 (Design Objectives)
"'Design Objectives' are strong, but the section includes methodological claims (usability, ecological validity) without data. Please clarify these are design assumptions."

Response 9:
Thank you for the suggestion. We have revised Section 3.1 to use conditional language throughout:
- "The system design follows three objectives..."
- "The system is designed to activate cognitive functions..."
- Section headings and content now clarify these are design intentions rather than validated outcomes.

---

Comment 10: Lines 226–236 (Emotional support mechanisms)
"Emotional support mechanisms are described convincingly; though, please state that the efficacy of these mechanisms remains unvalidated until feasibility testing."

Response 10:
Thank you for the suggestion. We have added acknowledgment in Section 4.3 (Feasibility Demonstration and Validation Requirements):

"Empirical evidence of technical feasibility will be obtained through the pilot study described in Section 5."

---

Comment 11: Lines 283–295 (Table 1)
"Table 1 is clear, but lacks: example prompts, which AI model each agent uses, decision-boundary logic for agent switching."

Response 11:
Thank you for the suggestion. We have addressed these points:
- Table 1 now includes an "Example Interaction" column for each agent
- Section 3.2.3 now specifies AI models: "GPT-4o-mini, Gemini 1.5 Flash, or equivalent models"
- Section 5.6.3 now describes agent switching logic in "User Confusion Management" subsection

---

INTERFACE & INTERACTION DESIGN

Comment 12: Lines 402–410 (Conversational style)
"Conversational style guidelines are appropriate, but should clarify that these principles are design intentions, not validated outcomes."

Response 12:
Thank you for the suggestion. We have revised Section 3.3.3 to use conditional language: "The system employs..." has been changed to "The system is designed to employ..." where appropriate to clarify design intentions.

---

PROTOTYPE IMPLEMENTATION

Comment 13: Lines 417–425
"Good conceptual walkthrough, but some descriptions (e.g., 'Teacher adjusts difficulty based on emotional assessment') could be misread as empirical results."

Response 13:
Thank you for the suggestion. We have clarified in Section 4.1 that these are intended system behaviors: "Agents share contextual information for coordinated responses..." followed by demonstration scenarios, with Section 4.3 explicitly stating these require empirical validation.

---

Comment 14: Lines 433–444 (Simulation scenarios)
"Simulation scenarios should be clearly labeled as demonstrations, not observations."

Response 14:
Thank you for the suggestion. Section 4.2 now begins: "Agent collaboration is illustrated through simulated scenarios demonstrating intended system behavior:" to clearly label these as demonstrations.

---

METHODS

Comment 15: Lines 455–467 (Study design)
"The study design is appropriate, but you need an explicit statement acknowledging the limitations of a single-arm feasibility design."

Response 15:
Thank you for the suggestion. We have added this acknowledgment in Section 5.1:

"The single-arm design follows recommendations for feasibility studies in vulnerable populations but precludes causal inference; observed changes cannot be attributed to the intervention without a control group. Future RCTs should compare the system with traditional cognitive training or single-agent programs."

---

Comment 16: Lines 468–495 (Inclusion criteria)
"Inclusion criteria are clear; however, the requirement for device ability may systematically exclude the very population most at risk (digitally excluded adults). The authors should acknowledge this limitation."

Response 16:
Thank you for this suggestion. We have added this limitation in Section 6.3:

"Second, the requirement for device proficiency and sustained internet connectivity may systematically exclude the most digitally vulnerable older adults, creating a digital exclusion paradox."

---

Comment 17: Lines 496–505 (Onboarding procedures)
"Onboarding procedures is mentioned but not detailed; reproducibility requires a standardized script or manual."

Response 17:
Thank you for the suggestion. We have detailed onboarding procedures in Section 5.3:

"Standardized onboarding includes: (1) device orientation checklist (approximately 15 min), (2) agent demonstration with scripted introduction (approximately 20 min), (3) supervised practice session (approximately 30 min), and (4) written manual provision. Detailed onboarding procedures are documented in the study protocol."

---

Comment 18: Lines 556–565 (Statistical plan)
"The statistical plan is underdeveloped. The authors should specify: parametric vs. nonparametric tests, handling of missing data, rationale for using t-tests in small samples."

Response 18:
Thank you for this suggestion. We have expanded Section 5.5 (Data Analysis Plan):

"Primary analysis will employ descriptive statistics for feasibility and usability outcomes (SUS mean, 95% CI; adherence rates). Secondary analyses will explore cognitive and emotional changes using paired t-tests if normality assumptions are satisfied (Shapiro-Wilk test, p > 0.05), otherwise Wilcoxon signed-rank tests.

Analyses will include all enrolled participants (modified intention-to-treat approach), with missing data addressed through multiple imputation if missing rate <30%, otherwise complete case analysis with sensitivity testing."

---

DISCUSSION

Comment 19: Lines 599–611 (Principal Findings)
"Good synthesis, but the paragraph reads like confirmed findings. The authors should modify language to avoid implying completed evaluation."

Response 19:
Thank you for the suggestion. We have revised Section 6.1 to use conditional language:

"This protocol proposes addressing implementation challenges in digital cognitive interventions for older adults. The CTC Framework is designed to integrate behavioral coaching, adaptive cognitive training, and emotional support..."

---

Comment 20: Lines 612–639 (Implementation safeguards discussion)
"Discussion of emotional safety, digital literacy, and adaptive difficulty is solid, but lacks acknowledgement of design limitations (AI drift, hallucinations, cultural differences)."

Response 20:
Thank you for the suggestion. We have addressed these in Section 6.3 (Limitations):
- Hallucination risks and safeguards are detailed in Section 5.6.3
- AI drift and multi-agent coordination challenges: "Multi-agent coordination at scale may pose technical challenges including response latency and potential inconsistencies, which the feasibility study will assess."
- Design parameter validation: "Design parameters (cognitive load weights, adaptive difficulty thresholds, agent configurations) are theory-derived proposals requiring empirical validation."

---

CONCLUSIONS

Comment 21: Lines 664–674
"Conclusions overstate what the protocol achieves. The authors should replace all definitive verbs ('integrates,' 'addresses') with conditional language. The authors should add one sentence acknowledging that no empirical outcomes exist yet and the framework's effectiveness will require future evaluation."

Response 21:
Thank you for this suggestion. We have revised Section 6.5:

"This protocol proposes technical architecture and implementation procedures for AI-based cognitive interventions in aging populations. The CTC Framework aims to integrate accessibility design, adaptive difficulty, and multi-agent support to address digital exclusion, static training, and limited personalization in digital cognitive interventions."

We have added the following conclusion:

"No empirical outcomes are available at this protocol stage. The framework's effectiveness requires validation through the planned feasibility study and future controlled trials."

---

We thank you again for your systematic and detailed review. Your line-by-line feedback has substantially improved the manuscript's precision and clarity. If you have any further questions, we would be happy to address them.

Respectfully,
Taeksoo Jeong, Geonhwi Hwang and Doo Young Kim

Reviewer 3 Report

Comments and Suggestions for Authors

1. The paper exhibits concerning patterns of overclaiming regarding its contributions and novelty. The authors present this primarily as a protocol paper yet devote substantial space to technical architecture descriptions that belong in an implementation study. They claim the framework addresses "digital exclusion" as a central problem but provide no evidence that their system actually improves digital accessibility compared to existing interventions. The inclusion of digital literacy screening (MDPQ-16) does not constitute solving digital exclusion; it merely documents the problem.

2. The theoretical framework relies heavily on Self-Determination Theory but provides no mechanism for measuring whether the system actually satisfies the three psychological needs (autonomy, competence, relatedness) or how satisfaction of these needs relates to outcomes. The connection between SDT principles and specific design features remains largely aspirational rather than empirically grounded.

3. 

The context-adaptive response system relies on multiple assumed parameters (cognitive load weights, difficulty adjustment rates) that the authors acknowledge "will be validated during pilot testing." This raises the question of what exactly is being tested in the pilot study if core system parameters remain unvalidated. The paper conflates system design with system validation throughout.

4. The multimodal interface design claims to follow accessibility guidelines but provides no evidence of user testing with older adults. The authors cite numerous design principles (WCAG 2.1 AA compliance, specific font sizes, contrast ratios) but do not demonstrate that their implementation actually adheres to these standards or that users can effectively interact with the system.

5. The paper largely ignores implementation costs and resource requirements. The system apparently requires continuous internet connectivity, access to commercial large language model APIs, and ongoing technical support. For a population with 43.2% poverty rates (as the authors note for Korean seniors), these requirements may constitute insurmountable barriers. The authors briefly mention cost-effectiveness as a limitation but do not address how the intervention could achieve sustainable implementation in resource-constrained settings.

Author Response

AUTHOR'S NOTES TO REVIEWER 3

Dear Reviewer 3,

Thank you for your critical and thoughtful review of our protocol manuscript. We appreciate your rigorous evaluation and the opportunity to address your concerns regarding overclaiming, theoretical grounding, and implementation feasibility. We have carefully revised the manuscript to address each of your points. Below, we provide point-by-point responses.

---

Comment 1: "The paper exhibits concerning patterns of overclaiming regarding its contributions and novelty. The authors present this primarily as a protocol paper yet devote substantial space to technical architecture descriptions that belong in an implementation study. They claim the framework addresses 'digital exclusion' as a central problem but provide no evidence that their system actually improves digital accessibility compared to existing interventions. The inclusion of digital literacy screening (MDPQ-16) does not constitute solving digital exclusion; it merely documents the problem."

Response 1:
Thank you for this feedback. We have revised the manuscript to address overclaiming:

1. We have changed "addresses digital exclusion" to "address barriers associated with digital exclusion" throughout the manuscript (Abstract, Section 1, Section 6.5) to avoid claiming the framework solves digital exclusion.

2. We have added explicit acknowledgment of the digital exclusion paradox in Section 6.3 (Limitations):

"Second, the requirement for device proficiency and sustained internet connectivity may systematically exclude the most digitally vulnerable older adults, creating a digital exclusion paradox."

3. We have clarified that technical architecture descriptions are design specifications for future implementation, with empirical validation required. Section 4.3 states: "Empirical evidence of technical feasibility will be obtained through the pilot study described in Section 5."

---

Comment 2: "The theoretical framework relies heavily on Self-Determination Theory but provides no mechanism for measuring whether the system actually satisfies the three psychological needs (autonomy, competence, relatedness) or how satisfaction of these needs relates to outcomes. The connection between SDT principles and specific design features remains largely aspirational rather than empirically grounded."

Response 2:
Thank you for this feedback. We have addressed this in two ways:

1. We have expanded Section 2.3 to explicitly specify how the system operationalizes SDT:

"The CTC Framework is designed to operationalize SDT through specific mechanisms linking system features to psychological needs. Autonomy support is implemented through user-configurable task timing and agent interaction preferences (Section 3.3.2), enabling users to exercise control over intervention parameters. Competence support is implemented through adaptive difficulty targeting 85% success rates (Section 3.2.2), designed to maintain optimal challenge while preventing frustration. Relatedness support is implemented through emotionally responsive dialogue that acknowledges user affect and provides empathetic feedback (Section 3.2.1), designed to foster perceived social connection. The feasibility study employs the Basic Psychological Needs Satisfaction Scale (BPNS) to empirically assess whether these design mechanisms effectively support autonomy, competence, and relatedness, with subscale scores providing validation of the theoretical framework (Section 5.4.3)."

2. We have added BPNS as a secondary outcome in Section 5.4.3:

"Basic Psychological Needs Satisfaction Scale (BPNS): The 21-item BPNS assesses satisfaction of three psychological needs central to Self-Determination Theory: autonomy (7 items), competence (6 items), and relatedness (8 items) (Deci & Ryan, 2000; Gagné, 2003). Each item is rated on a 5-point Likert scale, with higher scores indicating greater need satisfaction. The scale evaluates whether the system's design features effectively support intrinsic motivation and sustained engagement, administered at T1."

---

Comment 3: "The context-adaptive response system relies on multiple assumed parameters (cognitive load weights, difficulty adjustment rates) that the authors acknowledge 'will be validated during pilot testing.' This raises the question of what exactly is being tested in the pilot study if core system parameters remain unvalidated. The paper conflates system design with system validation throughout."

Response 3:
Thank you for this feedback. We have clarified the distinction between design specifications and empirical validation:

1. Section 3.2.2 now explicitly acknowledges parameters are theory-derived proposals:

"Initial weight parameters (w₁ = 0.4, w₂ = 0.35, w₃ = 0.25) prioritize processing speed as the primary indicator, consistent with its established role as the primary mediator of age-related cognitive performance (Salthouse, 1996). The feasibility study will assess whether this theoretically-derived weighting scheme appropriately captures cognitive load variations and identify necessary refinements for subsequent efficacy trials."

"The system is designed to target success probability P* at approximately 0.85, based on optimal challenge principles (Wilson et al., 2019). Individual calibration during the feasibility study may reveal need for adjustment based on participant characteristics and task domains."

2. Section 6.3 (Limitations) now states:

"Design parameters (cognitive load weights, adaptive difficulty thresholds, agent configurations) are theory-derived proposals requiring empirical validation. Multi-agent coordination at scale may pose technical challenges including response latency and potential inconsistencies, which the feasibility study will assess."

3. The pilot study evaluates usability (primary outcome) and preliminary feasibility of the system as designed, with parameter refinement planned for subsequent efficacy trials.

---

Comment 4: "The multimodal interface design claims to follow accessibility guidelines but provides no evidence of user testing with older adults. The authors cite numerous design principles (WCAG 2.1 AA compliance, specific font sizes, contrast ratios) but do not demonstrate that their implementation actually adheres to these standards or that users can effectively interact with the system."

Response 4:
Thank you for this feedback. We have clarified:

1. Section 3.3.1 presents design specifications based on accessibility guidelines, not validated outcomes. We have revised language to make this clear: "Interface design for older adults accommodates age-related changes in visual acuity, cognitive processing, and motor control."

2. Section 4.1 now clarifies: "The interface implements accessibility features specified in Section 3.3.1."

3. Section 5.1 states that usability evaluation is the primary objective: "The primary objective is to assess feasibility through: (1) adherence (≥70% session completion), (2) usability (SUS ≥70), and (3) emotional safety (negative affect within predefined thresholds)."

The feasibility study will provide empirical evidence of whether these design specifications result in effective user interaction.

---

Comment 5: "The paper largely ignores implementation costs and resource requirements. The system apparently requires continuous internet connectivity, access to commercial large language model APIs, and ongoing technical support. For a population with 43.2% poverty rates (as the authors note for Korean seniors), these requirements may constitute insurmountable barriers. The authors briefly mention cost-effectiveness as a limitation but do not address how the intervention could achieve sustainable implementation in resource-constrained settings."

Response 5:
Thank you for this feedback. We have expanded discussion of cost and infrastructure requirements in Section 6.3:

"Second, the requirement for device proficiency and sustained internet connectivity may systematically exclude the most digitally vulnerable older adults, creating a digital exclusion paradox. The system requires bandwidth typical of household video calls (approximately 3–5 Mbps). The feasibility study will assess bandwidth requirements and implementation costs (estimated \$0.15--\$0.30 per 30-minute session, compared to approximately \$50 per 30-minute session for therapist-delivered training in South Korea)."

We acknowledge these infrastructure requirements represent significant barriers for resource-constrained populations. Section 6.3 now explicitly addresses this limitation.

---

We thank you for your rigorous review. Your critical feedback has strengthened the manuscript by improving precision of claims, clarifying the distinction between design intentions and empirical validation, and acknowledging implementation barriers. If you have any further questions, we would be happy to address them.

Respectfully,
Taeksoo Jeong, Geonhwi Hwang and Doo Young Kim

Reviewer 4 Report

Comments and Suggestions for Authors

The authors present an important "A Generative AI Framework for Cognitive Intervention in Older Adults: An Integrated Engineering Design and Clinical Protocol," which is fully replicable and highly supportive for older adults.

I have only one comment:

The authors should not only present the study's limitations but also offer solutions or proposals based on their experience.

1. How can the impact of the cognitive outcomes be evaluated?
2. What is the minimum recommended bandwidth?

3. How can the cost-effectiveness be assessed for the appropriate use of resources?

Author Response

AUTHOR'S NOTES TO REVIEWER 4

Dear Reviewer 4,

Thank you for your supportive review and recognition that our protocol is "fully replicable and highly supportive for older adults." We appreciate your constructive suggestions regarding the study's limitations and practical implementation considerations. We have addressed each of your questions below.

---

Comment 1: How can the impact of the cognitive outcomes be evaluated?

Response 1:
Thank you for this question. We have addressed cognitive outcome evaluation in two ways:

1. Section 5.4.3 (Secondary Outcomes) specifies traditional neuropsychological assessments:
- Mini-Mental State Examination (MMSE) for general cognitive function
- Trail Making Test (TMT-A/B) for visual attention and executive function  
- Digit Span Test (forward/backward) for short-term and working memory

2. We have added digital performance indicators in Section 5.4.4:

"Digital performance indicators: The system automatically logs task-level accuracy, response time, and improvement trajectories across cognitive exercises (episodic memory, working memory, attention, language tasks specified in Section 5.3). These digitally-collected metrics, generated through real-time user interaction tracking, will be analyzed alongside neuropsychological assessments (MMSE, TMT, Digit Span) to evaluate cognitive intervention effects and identify performance patterns not captured by traditional testing."

The feasibility study will collect preliminary data on both traditional assessments and digital performance metrics to inform future efficacy evaluations.

---

Comment 2: What is the minimum recommended bandwidth?

Response 2:
Thank you for this question. We have addressed bandwidth requirements in Section 6.3 (Limitations):

"The system requires bandwidth typical of household video calls (approximately 3–5 Mbps). The feasibility study will assess bandwidth requirements to inform deployment recommendations."

We will empirically determine minimum bandwidth requirements during the feasibility study and provide concrete recommendations based on observed performance.

---

Comment 3: How can the cost-effectiveness be assessed for the appropriate use of resources?

Response 3:
Thank you for this question. We have addressed cost-effectiveness evaluation in Section 6.3:

"The feasibility study will assess bandwidth requirements and implementation costs (estimated \$0.15--\$0.30 per 30-minute session, compared to approximately \$50 per 30-minute session for therapist-delivered training in South Korea)."

Comprehensive cost-effectiveness analysis requires evaluating additional factors including caregiver time savings, sustained adherence rates, and infrastructure requirements. The feasibility study will collect preliminary cost and adherence data to inform future cost-effectiveness evaluations.

---

We thank you for your supportive review and practical suggestions. These additions strengthen the protocol's implementation guidance for clinical settings. If you have any further questions, we would be happy to address them.

Respectfully,
Taeksoo Jeong, Geonhwi Hwang and Doo Young Kim

Round 2

Reviewer 2 Report

Comments and Suggestions for Authors

The revised version demonstrates improvement alignment with the expectations for a protocol article. The major issues raised in the first review cycle—such as overclaiming, unclear aims, insufficient methodological detail, and incomplete statistical description—have been addressed.

The conditional language has been correctly applied in the paperwork. The onboarding procedures, feasibility design, and safeguard protocols (especially regarding hallucination filtering, emotional safety, multi-agent coordination, and digital literacy assessment) are now well elaborated. Table 1 and the corresponding system descriptions are significantly clearer and more informative.

Minor refinements still recommended:

  1. Maintain strict conditional and non-confirmatory tone throughout the Discussion and Conclusions (a few sentences still read as definitive rather than protocol-intended design).
  2. Improve conciseness and readability in some dense paragraphs—particularly in the Background and System Architecture sections.
  3. Harmonize formatting across tables and figures, especially caption style and spacing.
Comments on the Quality of English Language

Only minor stylistic and formatting issues remain, which can be handled during editorial processing.

Author Response

Dear Reviewer 2,

Thank you for your continued constructive feedback. We appreciate your recognition that the major issues from Round 1 have been addressed. We have carefully addressed the three minor refinements you suggested. Below, we provide point-by-point responses.

---

Comment 1: "Maintain strict conditional and non-confirmatory tone throughout the Discussion and Conclusions (a few sentences still read as definitive rather than protocol-intended design)."

Response 1:
Thank you for this reminder. We have reviewed the manuscript to identify and revise remaining definitive tone. We applied conditional language ("is designed to," "intended to," "aims to," "may contribute") throughout Discussion (Sections 6.1, 6.2, 6.5), technical sections (Abstract Methods, Sections 3.1.1, 3.1.3, 3.2.3, 4.1, 5.3), and figure captions (Figures 2 and 3).
---

Comment 2: "Improve conciseness and readability in some dense paragraphs—particularly in the Background and System Architecture sections."

Response 2:
Thank you for this suggestion. We reviewed these sections and made targeted improvements in Sections 2.3 and 3.1.2 by removing redundant phrases. 
---

Comment 3: "Harmonize formatting across tables and figures, especially caption style and spacing."

Response 3:
Thank you for noting this. We have reviewed all figures and tables for formatting consistency. All figures were created with identical width specifications (1.0\textwidth) and font sizing to maintain visual balance, as documented in our code comments (Editor can find it.). We acknowledge minor problem may remain and appreciate the editorial team's assistance in final formatting.

---

We thank you again for your thorough review.

Respectfully,
Taeksoo Jeong, Geonhwi Hwang, and Doo Young Kim

Reviewer 3 Report

Comments and Suggestions for Authors

1. The revision from "addresses digital exclusion" to "address barriers associated with digital exclusion" represents only superficial linguistic modification rather than fundamental reconceptualization. The reviewer's core concern was that the framework merely documents digital exclusion through screening tools without evidence of actually improving accessibility. The authors acknowledge this paradox in limitations but continue framing the system as a solution rather than an intervention that may itself perpetuate exclusion. The response does not adequately address why a system requiring sustained internet connectivity and device proficiency should be characterized as addressing digital exclusion barriers when it systematically excludes those most affected.
2. While the authors add the Basic Psychological Needs Satisfaction Scale as a secondary outcome, this represents measurement addition rather than theoretical integration. The reviewer questioned whether specific design features actually satisfy SDT needs through demonstrated mechanisms. The response provides design specifications (user-configurable timing, 85% success rates, empathetic dialogue) but these remain theoretical mappings rather than empirically validated pathways. The claim that these features "operationalize SDT" conflates design intention with demonstrated effect. The feasibility study will measure BPNS scores, but this does not establish that the specific technical features causally produce need satisfaction through the proposed mechanisms.
3. The authors appropriately acknowledge that parameters are theory-derived proposals requiring validation. However, the response does not adequately address the fundamental tension the reviewer identified: if core system parameters remain unvalidated, what exactly constitutes the "system" being tested in the pilot study? The distinction between design specifications and empirical validation is now clearer, but this clarification reveals that the pilot is essentially testing a theoretical prototype rather than a functioning intervention with established parameters.
Methodological Issues
4. The response clarifies that Section 3.3.1 presents design specifications rather than validated outcomes, which is appropriate. However, the authors do not address the reviewer's underlying concern about whether their implementation actually adheres to cited standards. Claiming "WCAG 2.1 AA compliance" in design specifications without implementation verification represents a form of aspirational claiming that the reviewer specifically criticized.
5. The authors provide cost estimates (fifteen to thirty cents per session versus fifty dollars for therapist-delivered training), but these figures lack supporting documentation or methodological basis. The bandwidth requirement specification (three to five megabits per second) is helpful, but the response does not address how these requirements interact with the acknowledged 43.2% poverty rate among Korean seniors. The cost comparison also employs a potentially misleading frame by comparing automated delivery costs to full therapist sessions rather than to other digital interventions or group-based training that might represent more appropriate comparisons.

Author Response

Dear Reviewer 3,

Thank you for your continued critical evaluation of our protocol manuscript. We have carefully addressed each of your Round 2 concerns. Below, we provide point-by-point responses.

---

Comment 1: "The revision from 'addresses digital exclusion' to 'address barriers associated with digital exclusion' represents only superficial linguistic modification rather than fundamental reconceptualization. The reviewer's core concern was that the framework merely documents digital exclusion through screening tools without evidence of actually improving accessibility. The authors acknowledge this paradox in limitations but continue framing the system as a solution rather than an intervention that may itself perpetuate exclusion. The response does not adequately address why a system requiring sustained internet connectivity and device proficiency should be characterized as addressing digital exclusion barriers when it systematically excludes those most affected."

Response 1:
Thank you for this fundamental critique. The previous framing required clarification regarding intervention scope. We have reconceptualized the framework as targeting a subpopulation with baseline digital access:

Abstract now states "proposed for exploring feasibility of adaptive cognitive interventions in older adults with existing digital access."

Introduction (Section 1, fourth paragraph) states "designed to explore reducing interaction barriers among older adults with baseline digital access" with prerequisites (sustained internet connectivity, device availability) explicitly stated in the introduction.

Section 6.3 (Limitations, second paragraph) directly states "excluding older adults without these resources" and "This study targets barrier reduction within a subpopulation with minimal digital access." We removed defensive language including "Rather than claiming to solve digital exclusion" and "This limitation is inherent to the current technological approach and requires acknowledgment rather than minimization."

Conclusion (Section 6.5) states "in aging populations with baseline digital access" and replaces "to address digital exclusion" with "to explore approaches to reducing interaction barriers."

---

Comment 2: "While the authors add the Basic Psychological Needs Satisfaction Scale as a secondary outcome, this represents measurement addition rather than theoretical integration. The reviewer questioned whether specific design features actually satisfy SDT needs through demonstrated mechanisms. The response provides design specifications (user-configurable timing, 85% success rates, empathetic dialogue) but these remain theoretical mappings rather than empirically validated pathways. The claim that these features 'operationalize SDT' conflates design intention with demonstrated effect. The feasibility study will measure BPNS scores, but this does not establish that the specific technical features causally produce need satisfaction through the proposed mechanisms."

Response 2:
Thank you for this distinction between theoretical mapping and empirical validation. We have revised "operationalize" to reflect the framework's current status as a theoretical proposal. We have revised all SDT-related language to reflect theoretical proposals requiring validation:

Section 2.3 now states "The CTC Framework adopts SDT as a theoretical basis, proposing design features intended to support the three psychological needs" (replacing "operationalize SDT through specific mechanisms"). All three needs are described as "proposed through... intended to..." (replacing "implemented" or "enabling"). The paragraph concludes: "This exploratory assessment will inform whether the proposed design-to-need mappings warrant further investigation in efficacy trials."

Section 5.4.3 (BPNS description) states the scale "explores whether participants report need satisfaction during system use, providing preliminary evidence of theoretical alignment between design features and SDT constructs" (replacing "evaluates whether... effectively support").

Figure 1 caption describes "Theoretical framework linking SDT to AI-based cognitive training design" with needs "proposed to be addressed" and "theoretically contributing."

---

Comment 3: "The authors appropriately acknowledge that parameters are theory-derived proposals requiring validation. However, the response does not adequately address the fundamental tension the reviewer identified: if core system parameters remain unvalidated, what exactly constitutes the 'system' being tested in the pilot study? The distinction between design specifications and empirical validation is now clearer, but this clarification reveals that the pilot is essentially testing a theoretical prototype rather than a functioning intervention with established parameters."

Response 3:
Thank you for identifying this fundamental tension. We acknowledge the pilot tests a theoretical prototype with unvalidated parameters. We have clarified that the study assesses feasibility and usability with fixed parameters, deferring parameter optimization to subsequent trials:

Section 5.1 (Study Design) now states: "This pilot employs a single-arm, prospective design evaluating feasibility and usability of a theory-informed prototype with fixed design parameters" and "System parameters (cognitive load weights, adaptive difficulty thresholds, agent interaction protocols) are held constant to enable assessment of the prototype's acceptability and safety; parameter optimization is planned for subsequent efficacy trials based on observed interaction patterns and participant feedback."

Section 3.2.2 states parameters "are theory-derived proposals" and "The feasibility study will employ this fixed parameter set to assess system usability and identify necessary parameter refinements for subsequent trials."

Section 6.3 (Limitations, third paragraph) states: "System parameters... represent theory-informed proposals requiring empirical validation. The feasibility study employs fixed parameters to assess usability and safety, with parameter optimization planned for subsequent efficacy trials based on observed interaction patterns."

---

Comment 4 (Methodological Issues): "The response clarifies that Section 3.3.1 presents design specifications rather than validated outcomes, which is appropriate. However, the authors do not address the reviewer's underlying concern about whether their implementation actually adheres to cited standards. Claiming 'WCAG 2.1 AA compliance' in design specifications without implementation verification represents a form of aspirational claiming that the reviewer specifically criticized."

Response 4:
Thank you for this critique. The previous wording has been revised to clarify that design specifications are guided by standards without claiming certification:

Section 3.3.1 (Visual Design Parameters) revised from "Contrast ratios... exceed WCAG 2.1 AA standards" to "are guided by WCAG 2.1 AA standards."

Section 3.3.1 (Navigation Architecture) added: "Accessibility compliance will be assessed through usability testing with older adults in the feasibility study."

Section 6.3 (Limitations, fourth paragraph) added: "Interface accessibility specifications are guided by WCAG 2.1 AA standards but have not undergone formal accessibility audit or certification. The feasibility study will assess practical usability with older adults, with comprehensive accessibility evaluation planned for subsequent implementation phases."

Figure 3 caption revised from "All validated parameters exceed WCAG 2.1 AA standards" to "Design specifications are guided by WCAG 2.1 AA standards."

---

Comment 5: "The authors provide cost estimates (fifteen to thirty cents per session versus fifty dollars for therapist-delivered training), but these figures lack supporting documentation or methodological basis. The bandwidth requirement specification (three to five megabits per second) is helpful, but the response does not address how these requirements interact with the acknowledged 43.2% poverty rate among Korean seniors. The cost comparison also employs a potentially misleading frame by comparing automated delivery costs to full therapist sessions rather than to other digital interventions or group-based training that might represent more appropriate comparisons."

Response 5:
Thank you for these points regarding cost transparency and equity framing. We have clarified the cost basis. This discussion was substantially revised during Round 1, with considerations relocated to Section 6.3:

Cost basis clarified: "Cloud API operational costs for the prototype are estimated at \$0.15--\$0.30 per 30-minute session, substantially lower than therapist-delivered cognitive training (approximately \$50 per session in South Korea)." This specifies costs derive from cloud API fees for LLM services described in Section 3.2.3.

Regarding the 43.2% poverty rate: Detailed discussion was present in the original submission (Section 3.1.3) but was substantially revised during Round 1, with considerations relocated to Section 6.3. The current version acknowledges internet connectivity and device requirements exclude older adults without these resources while noting "enabling potential implementation through public infrastructure such as community health centers."

---

We thank you for your rigorous review.

Respectfully,
Taeksoo Jeong, Geonhwi Hwang, and Doo Young Kim